

# Global and regional Pleistocene benthic δ¹⁸O stacks with a comparison of different age modeling strategies

Yuxin Zhou[1], Lorraine E. Lisiecki[2], Stephen R. Meyers[3], Taehee Lee[4], Charles Lawrence[4]

[1]Department of Earth and Atmospheric Sciences, Georgia Institute of Technology, Atlanta, GA 30332, USA
[2]Department of Earth Science, University of California, Santa Barbara, Santa Barbara, CA 93106, USA
[3]Department of Geoscience, University of Wisconsin–Madison, Madison, WI 53706, USA
[4]Department of Applied Mathematics, Brown University, Providence, RI 02912, USA

*Correspondence to*: Lorraine E. Liseicki (lisiecki@geol.ucsb.edu)

**Abstract.** Constructing accurate age models for Pleistocene marine sediments is crucial for our understanding of glacial-
interglacial cycles and other climatic processes. Benthic foraminiferal δ¹⁸O stacks, a proxy for ice sheet and climate evolution, are often used for stratigraphic alignment and chronology development in deep-sea sedimentary records, in combination with biostratigraphy, paleomagnetism, and radioisotopic constraints. Selection of an appropriate benthic δ¹⁸O alignment target influences the derived chronology at a given site, and divergent regional trends in benthic δ¹⁸O highlight the need for ocean-specific benthic δ¹⁸O stacks. The specific scientific question to be addressed by a study may also influence whether the
alignment target should include astronomical tuning. Here, we introduce three benthic δ¹⁸O stacks – Atlantic, Pacific, and global – with three distinct chronologies for the global stack that incorporate astronomical forcing constraints to various degrees. The new global stack utilizes data from 224 cores and includes 50% more data than the previous "ProbStack" (Ahn et al., 2017). Hand-tuned regional and global stacks, intended as updates to the "LR04" stack (Lisiecki and Raymo, 2005), incorporate chronologies transferred from absolutely dated archives during 0-654 thousand years ago (ka) and an
astronomically forced ice sheet model during 654-2700 ka. Due to the heterogeneous nature of the age constraints used for these stacks, we call them BIGSTACK$_{mixed}$, BIGSTACK$_{mixedA}$, and BIGSTACK$_{mixedP}$. For applications where astronomical tuning should be minimized, we present a global stack primarily constrained by geomagnetic reversal age estimates, BIGSTACK$_{magrev}$. We also develop a third age model, BIGSTACK$_{auto}$, which uses an automated optimization algorithm to "minimally tune" the stack to the pervasive ~41 kyr obliquity cycle, while avoiding assumptions about astronomical phase
relationships. This suite of stacks offers flexibility in choosing δ¹⁸O stratigraphic alignment targets, to allow a wide range of applications in paleoceanographic hypothesis testing.

## 1 Introduction

The stable oxygen isotope signature (δ¹⁸O) of benthic foraminifera has long been used to study paleoclimate change. Benthic foraminiferal δ¹⁸O stacks – aggregated benthic δ¹⁸O records designed to maximize the signal-to-noise ratio using data from
multiple cores – are important benchmarks for the stratigraphic alignment of deep-sea sedimentary records beyond the range of radiocarbon (>55 thousand years or kyrs). Stratigraphic alignment based on benthic δ¹⁸O stacks can synchronize records



from different regions under the assumption that δ¹⁸O varies nearly synchronously (i.e., within ~2 kyr (Rand et al., 2024)) throughout the deep sea. Sedimentary records need to be put on a common reference timeline so that the leads and lags of climatic processes, as well as claims of causality, can be scrutinized. In addition to stratigraphy, benthic δ¹⁸O itself offers a first-order characterization of ice sheet and deep-sea temperature evolution in response to forcing (Shackleton and Opdyke, 1973; Huybers and Wunsch, 2004; Lisiecki and Raymo, 2007; Rohling et al., 2009).

The 5.3-million-year-long Pliocene-Pleistocene LR04 benthic δ¹⁸O stack (Lisiecki and Raymo, 2005) is widely used by the community. In recent years, regional trends inconsistent with the LR04 global stack have been identified (Lisiecki and Stern, 2016; Wilkens et al., 2017; Caballero-Gill et al., 2019; Zhou et al., 2024). Eight regional δ¹⁸O stacks (Lisiecki and Stern, 2016) address the spatial deviations in benthic δ¹⁸O but only cover the last glacial cycle (0-150 ka). LR04's Pliocene-Pleistocene stack successor, "ProbStack", gathered more benthic δ¹⁸O records (180 vs. 57 in LR04) and introduced an uncertainty analysis generated with a profile Hidden Markov Model (Ahn et al., 2017). However, ProbStack used LR04 as the initial target for stack construction. As a result, ProbStack's age information largely inherits that of the LR04 stack and does not address chronological inaccuracies that have been identified in LR04. Like LR04, ProbStack does not include regional stacks and thus lacks support for localized stratigraphic alignment.

In this study, we leverage an increased number of published benthic δ¹⁸O records and the recently developed BIGMACs algorithm (Lee et al., 2023) to introduce three Pleistocene benthic δ¹⁸O stacks – Atlantic, Pacific, and global – with three different age models for the global stack. A global stack, BIGSTACK$_{mixed}$, is intended as an update to the LR04 global stack and uses speleothem-based age constraints from 0-654 ka and tuning to an ice sheet model beyond. Region-specific alignment targets following the same age model strategy are provided by Pacific and Atlantic stacks, BIGSTACK$_{mixedP}$ and BIGSTACK$_{mixedA}$. For applications where astronomical tuning should be minimized, we present an age model for the global stack, BIGSTACK$_{magrev}$, with age estimates derived primarily from paleomagnetic reversal ages (Ogg, 2020) and the constraint of stabilizing global sedimentation rates. Additionally, an auto-tuned global stack age model, BIGSTACK$_{auto}$, is generated using an optimization algorithm to "minimally tune" to the pervasive ~41 kyr obliquity cycle, while avoiding assumptions about astronomical phase relationships. We do not present Pliocene stacks as part of the present study, as our preliminary analyses indicate that the reduced number of records and the weaker signal-to-noise ratio in Pliocene benthic δ¹⁸O generate large alignment uncertainties that impede the effectiveness of our stacking method.

## 2. Background

Constructing an age model for Pleistocene sediments presents various challenges. Existing methods differ in the age ranges over which they can be applied, assumptions, and levels of uncertainty. For the sake of this work, we consider three categories of dating methods: radioisotopically dated marine strata and events (e.g., biostratigraphic and paleomagnetic events), varve



counting and astronomical tuning, and correlation to radioisotopically dated terrestrial archives (e.g., speleothems and ice
cores). We briefly review the strengths and weaknesses of each category, as relevant to constructing multiple versions of the
benthic $\delta^{18}$O stacks.

## 2.1 Radioisotopically dated marine strata and events

Methods that assign absolute ages using radioisotopic data provide powerful constraints on chronologies. Radiocarbon dating,
for example, can date sediments up to 55 ka in age (Heaton et al., 2020), provided that planktic foraminifera shells or plant
fragments are present. At sites that receive volcanic ash of known ages, tephrochronology may be established (Lowe, 2011).
Differentiating ash layers geochemically and linking them to specific eruptive events remains the main challenge of this method
(Davies et al., 2014). Lastly, magnetostratigraphy (Ogg, 2020) and biostratigraphy (Berggren and Van Couvering, 2011) offer
absolute ages when radioisotopic dating on magnetic reversal or first/last appearance datum is possible. However, these
horizons do not always coincide with materials suitable for absolute dating, and they are sometimes dated by astronomical
tuning instead, adding another layer of age uncertainty. Magnetostratigraphic and biostratigraphic horizons, even if present
and preserved, are also often hundreds of thousands of years apart, leaving long temporal gaps in the age model constraints.
Diachroneity in biostratigraphy due to paleoecological changes (Zimmerman et al., 2025), taxonomic inconsistencies (Lam et
al., 2022), or a number of other factors can complicate its chronological interpretation.

## 2.2 Astronomical tuning

At sites such as the Santa Barbara Basin and Cariaco Basin with varve preservation, an age model can be established by varve
counting (Schimmelmann et al., 2013; Hughen and Heaton, 2020), although varve preservation in sediments is rare. However,
on longer timescales, regular sedimentary alternations associated with precession, obliquity, and eccentricity are routinely used
to develop chronologies via astronomical tuning. Astronomical tuning has the potential to "fill in the blanks" between absolute
ages that can be far apart and produce a detailed age model accurate to within several kiloyears. The tuning process is versatile;
a variety of records, including oxygen and carbon stable isotopes, color reflectance, magnetic susceptibility, gamma ray, and
X-ray fluorescence elemental abundance data, have been used for tuning (Tiedemann and Haug, 1992; Shackleton, 1997;
Westerhold et al., 2007; Meyers, 2015, 2019; Ma et al., 2023). Similarly, tuning targets have varied from study to study, with
the common ones being orbital frequencies, insolation curves (e.g., Laskar et al., 2004), ice models (e.g., Imbrie and Imbrie,
1980), and domain-specific tuned reference datasets (e.g., Lisiecki and Raymo, 2005). Astronomical tuning can be done
manually or using statistical algorithms (e.g., Martinson et al., 1982; Malinverno et al., 2010; Meyers, 2015, 2019; Li et al.,
2019). Astronomical tuning can be prone to false positives (detecting astronomical signals where they do not exist) and false
negatives (failure to detect astronomical signals) (Vaughan et al., 2011; Hilgen et al., 2015; Waltham, 2015; Kemp, 2016;
Sinnesael et al., 2019). While false negatives lead to missed opportunities to create accurate age models, the danger of false
positives can be considered greater in that they introduce erroneous chronological constraints.






In the case of the late Pleistocene 100-kyr cycles, debate surrounds the origin of the cycles and can complicate the tuning attempts (Huybers and Wunsch, 2005; Abe-Ouchi et al., 2013; Barker et al., 2025). Tuning to precession requires choosing whether the NH or SH precession signal is the most appropriate target because, unlike obliquity, precession summer insolation forcing is antiphased between the hemispheres. A mechanistic understanding of the seasonal and latitudinal propagation of

precessional signals to sedimentary records is thus required. Astronomical tuning is also limited by the validity of the theoretical astronomical solutions. Beyond 10 Ma, the Earth's obliquity and precession are not well constrained (Zeeden et al., 2014). Accurate solutions for eccentricity stretch to 50-55 Ma, beyond which the solar system simulations are impacted by chaos, precluding an accurate reconstruction (Laskar et al., 2011). Astronomical age models produced without direct tuning to the theoretical solution are considered floating timescales, but they can be anchored by absolute ages when available

(radioisotopic dating, paleomagnetic reversals, etc.). Lastly, only records with low levels of noise and of sufficient temporal span and resolution are suitable for astronomical tuning.

## 2.3 Correlation to radioisotopically-dated terrestrial archives

Speleothem records are frequently high in temporal resolution and can be radioisotopically dated, offering an attractive target for aligning marine sediments. This technique requires marine records that can be reasonably associated with terrestrial

hydroclimatic signals from speleothems. Previous applications invoked the synchronicity between the mid-latitude North Atlantic sea surface temperature and Mediterranean rainfall (Drysdale et al., 2009; Govin et al., 2015), North Atlantic ice rafting and Asian monsoon strength (Lisiecki and Stern, 2016), or South China Sea temperature/salinity and Asian monsoon dynamics (Caballero-Gill et al., 2012). Well dated as the speleothems may be, they are primarily found in tropical and mid-latitude regions, and records older than 650 ka (the limit of U-Th dating) are rare (Cheng et al., 2016; Engel and Pickering,

2022). The application of U-Pb dating to speleothems has the potential to alleviate the limited age range and offer more opportunities for speleothem-marine correlation (Woodhead and Pickering, 2012). The lack of modeling studies demonstrating a firm mechanistic basis for correlation between the desired land-sea link and lead/lag estimates often adds uncertainty to this technique.

Ice core data provide another option for mechanistically linking sedimentary records to absolute ages, typically based on the co-variation of polar/subpolar sea surface temperature (SST) and polar air temperature in Greenland (Bond et al., 1993) and Antarctica (Lamy et al., 2004). Several studies have constructed ice core-based marine age models (e.g., Govin et al., 2009, 2012; Martrat et al., 2007). Marine lithogenic flux data have also been correlated with Antarctic dust records (Anderson et al., 2014). This approach is limited by how far back ice cores reach – the Antarctic ice core dates back to 800 ka (EPICA

Community Members, 2004; Jouzel et al., 2007), although this is somewhat alleviated by a new core reportedly extending back to 1.2 Ma, while the oldest the Greenland ice core can reach is 129 ka (NEEM community members, 2013). Ice cores drill sites present another limitation – sediments from lower latitudes are a long distance away and are mostly not suitable to use this chronostratigraphic correlation approach. Importantly for this study, we note that the Antarctic ice core chronologies





(AICC) 2012 and 2023 use astronomical tuning in between absolutely dated depths (Bazin et al., 2013; Bouchet et al., 2023).

Studies that wish to use untuned marine records should thus avoid correlating SST or marine lithogenic flux to Antarctic ice core data on the AICC scales.

## 2.4 Age interpolation between tie points

The aforementioned methods typically provide age-depth "tie points" (or "control points") for marine records with gaps in between, necessitating interpolation between depths of known ages. Linear age interpolation is the most simplistic approach,

with the assumption that sedimentation rate stays constant between any given set of tie points. At individual sites, this assumption may not be realistic because of local changes in preservation, export productivity, circulation, ice-rafting, dust flux, etc. that can cause abrupt (< 100 years) changes in the sediment accumulation (e.g., McManus et al., 1998). However, the observation of sudden, large-amplitude sediment accumulation rate changes are unlikely in slowly accumulating deep ocean sediments, due to processes that rework the sediment, such as bioturbation, which tend to smooth over accumulation

"kinks." Although long-term (> 1000 years) changes in the global sediment accumulation are expected due to climate-driven changes in terrestrial weathering, ocean productivity, and deep-sea carbonate dissolution (Cartapanis et al., 2016, 2018; Kienast et al., 2016; Costa et al., 2020), sediment coring sites used in the present study are selected for relatively steady sedimentation rate through time. These expectations form the basis of many age-model algorithms that evaluate potential age-depth relations and penalize those that produce large variability in sedimentation rate (Brüggemann, 1992; Blaauw and Christen, 2011; Lin et

al., 2014; Lee et al., 2023). Linear interpolation between tie points plays an important role in constructing untuned stacks that seek to avoid assuming the input records are astronomically forced. Untuned stacks (e.g., Huybers and Wunsch, 2004; Lisiecki, 2010) typically minimize the changes in the average sedimentation rate across globally distributed core sites using magnetic reversal ages as age control points and correcting for downcore sediment compaction.

## 3 Methods

### 3.1 Stacking

As a first step, we use the software package Bayesian Inference Gaussian Process regression and Multiproxy Alignment for Continuous Stacks (BIGMACS) (Lee et al., 2023) to create a Pleistocene global benthic $\delta^{18}O$ stack. The BIGMACS software constructs a stack from ocean sediment core data by iteratively creating multiproxy age models, first using an initial alignment target, and then building and updating the stack using Gaussian process regression (Williams and Rasmussen, 2006) of the

data on their multiproxy age models. Here we use 224 benthic $\delta^{18}O$ records (Supplementary Data File 1; Fig. 1) compiled by Zhou et al. (2024), which increase the number of $\delta^{18}O$ measurements by 50% compared to ProbStack (Ahn et al., 2017). We





name the resulting stack BIGSTACK$_{LR04}$ to reflect that it is a global stack based on the LR04 stack age model (Fig. S1).

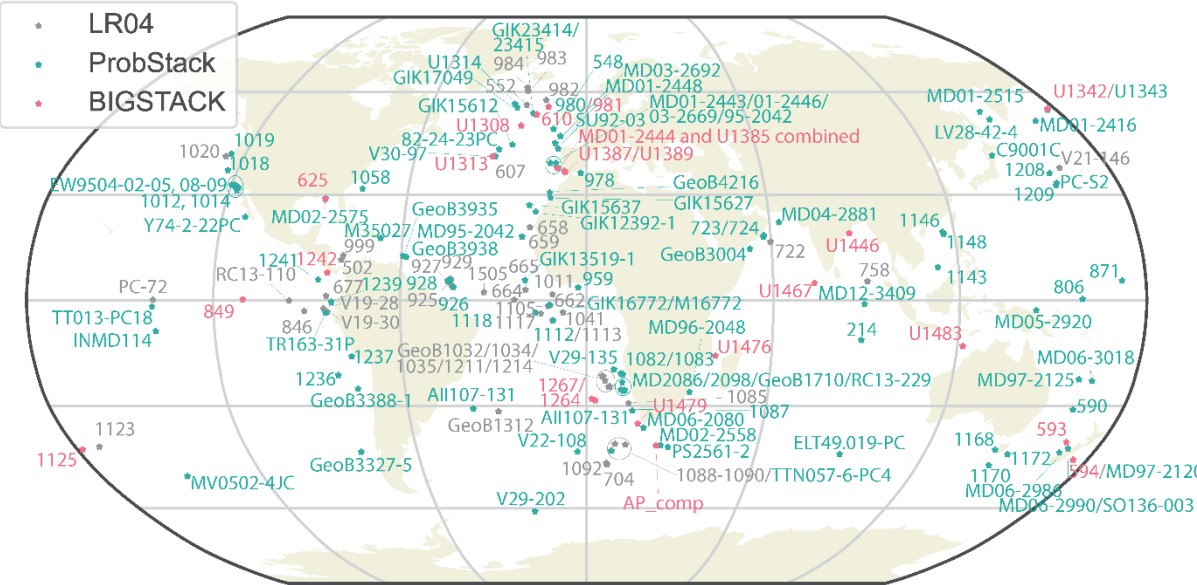

**Figure 1. Map of the input cores for BIGSTACKs, including existing compilations and additional records newly compiled for this**
**study.**

Although we use the LR04 stack as the initial alignment target, we update the global stack age model using magnetic reversal

ages and tuning, as described in sections 3.2 and 3.3 (Fig. 2).

Because benthic $\delta^{18}$O records can differ in timing and amplitude regionally (Lisiecki and Stern, 2016; Rand et al., 2024; Zhou

et al., 2024), we also use BIGMACS (Lee et al., 2023) to separately construct Atlantic and Pacific stacks using 125 Atlantic

records and 81 Pacific benthic $\delta^{18}$O records, respectively (Fig. 2). We call these stacks BIGSTACK$_{mixedA}$ and BIGSTACK$_{mixedP}$.

From 0-654 ka, the initial alignment targets used by BIGMACS are constructed regionally from the LS16 regional stacks

(Lisiecki and Stern, 2016) and the H23NA North Atlantic stack (Hobart et al., 2023), both of which are untuned. Because the

H23NA stack and the LS16 regional stacks have different amplitudes and a mean offset, we calculated a linear best fit between

them and adjusted the scaling of the LS16 regional stacks to match that of the H23NA stack before joining the two stacks to

create our alignment target. The following equations were used to scale and offset the LS16 regional stacks to match that of

the H23NA stack for our alignment targets:

$$DNA\_scaled\_to\_H23NA = 0.91*DNA - 0.08$$

$$DP\_scaled\_to\_H23NA = 1.12*DP - 1.04$$



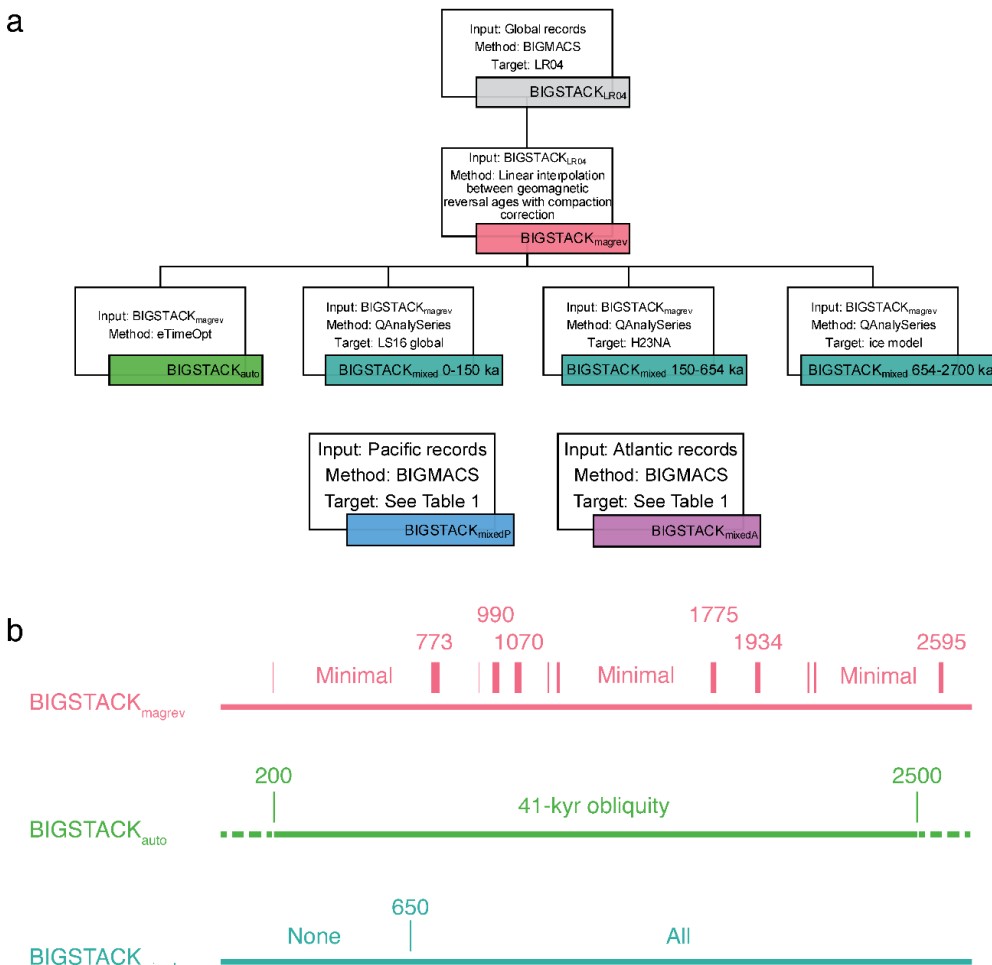

**Figure 2. Stack version differences. (a)** A flow chart showing the creation process of the different stack versions. The white boxes contain the construction strategy, and the colored boxes contain the name of the resulting stack. The connecting lines denote the stack construction sequence. **(b)** Astronomically based age information applied in each stack. Numbers are in thousands of years ago. In BIGSTACK$_{magrev}$, the tick marks denote the geomagnetic reversals and excursions, and the thickness of the tick marks is proportional to the number of cores within which the reversals and excursions were identified. In BIGSTACK$_{auto}$, the first and last 200-kyr segments are from BIGSTACK$_{mixed}$. BIGSTACK$_{mixedP}$ and BIGSTACK$_{mixedA}$ use the same tuning information as BIGSTACK$_{mixed}$.

where DNA and DP are the LS16 deep North Atlantic and deep Pacific stacks, respectively. The splice point between them was selected where the two stacks are in good agreement, at 136 ka. To account for different deep water residence times, we add a 1-kyr lag to H23NA when creating the Pacific regional stack target (Stern and Lisiecki, 2014); the uncertainty associated this lag is discussed in section 5.1. Based on sea-level reconstructions (Barnett et al., 2023; Dumitru et al., 2023), we also stretch the last interglacial in our alignment target by shifting the end of Marine Isotope Stage (MIS) 5e from 118 ka



| Chronozone | Age (ka) |
|---|---|
| Core top | 0 |
| Iceland Basin Event | 188 |
| Brunhes/Matuyama | 773 |
| Kamikatsura | 885 |
| Santa Rosa | 930 |
| Top Jaramillo | 990 |
| Base Jaramillo | 1070 |
| Punaruu | 1125 |
| Top Cobb Mountain | 1180 |
| Base Cobb Mountain | 1215 |
| Top Olduvai | 1775 |
| Base Olduvai | 1934 |
| Top Reunion | 2116 |
| Base Reunion | 2140 |
| Matuyama/Gauss | 2595 |
| Top Kaena | 3032 |
| Base Kaena | 3116 |
| Top Mammoth | 3207 |
| Gauss/Gilbert | 3596 |

**Table 1. Geomagnetic reversal and excursion ages used to construct the BIGMACS$_{magrev}$, obtained from Geological Time Scale 2020**
**(Ogg, 2020).**

to 115 ka, to match updated estimates for the age of glacial inception. However, through the iterative stacking approach of
BIGMACS, the resulting stack is not forced to maintain the same age constraints.

Beyond 654 ka, both regional stacks were constructed using the tuned global stack BIGSTACK$_{mixed}$ (as described in section
3.4) as their initial alignment target, with a regional adjustment to the alignment targets from 1.8-1.9 Ma. Zhou et al. (2024)
found that differences between Atlantic and Pacific stacks are relatively small from 2700 ka to 654 ka, except during 1.8-1.9
Ma. We thus spliced in the regional stacks from that study for 1.8-1.9 Ma (Zhou et al., 2024) to BIGSTACK$_{mixed}$ for the
regional alignment targets and directly into the resulting regional stacks, BIGSTACK$_{mixedA}$ and BIGSTACK$_{mixedP}$. Following
Zhou et al. (2024), MIS 64 and 74 are aligned to the obliquity minima at 1.793 and 1.958 Ma in both stacks; in only the Atlantic
stack, we additionally anchor MIS 68 and 70 to Northern Hemisphere summer insolation minima.



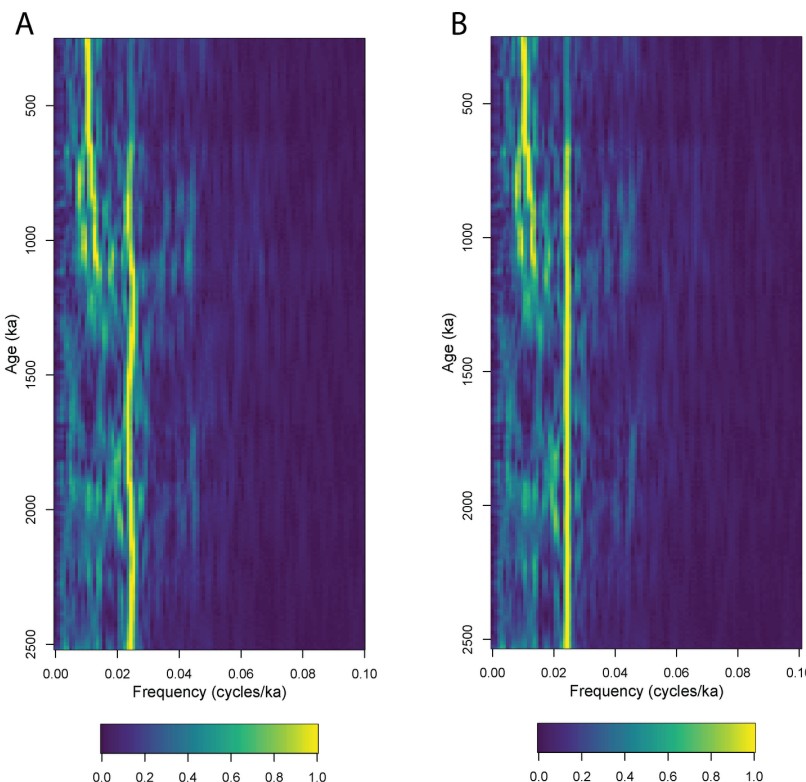

**Figure 3. Time-frequency analysis results from multi-taper method evolutive harmonic analysis of the untuned δ¹⁸O stack (A) and the auto-tuned δ¹⁸O stack after the application of eTimeOpt (Meyers, 2019) (B). The analysis utilizes a 400-kyr moving window with three 2π prolate tapers. A linear trend has been removed from each window, and the maximum amplitude for each window is normalized to unity.**


## 3.2 Magnetic reversal-constrained age model

To construct a benthic $\delta^{18}O$ stack age model where astronomical tuning is minimized, we compile 35 cores with both benthic $\delta^{18}O$ and paleomagnetic measurements (Supplementary Data File 2). We then assume that the core's sedimentation rate is constant between the bracketing depths dated with geomagnetic reversals, after applying a correction for sediment compaction.

Compaction from the weight of the overlying sediment will systematically generate the appearance of slower sediment accumulation at greater core depths (Velde, 1996). Therefore, we apply a correction for sediment compaction similar to Lisiecki (2010), which derived a relationship between porosity and depth based on a compilation of eight sediment cores (Huybers and Wunsch, 2004). Thus, we approximate sediment porosity as $\Phi=70+10\times e^{-0.02d}$, where $\Phi$ is the porosity and d is the core depth in meters. Additionally, we fix the core top to be 0 ka so that the uppermost segment of the core can be assigned

ages, although we acknowledge that the core top sediments need not always be modern.



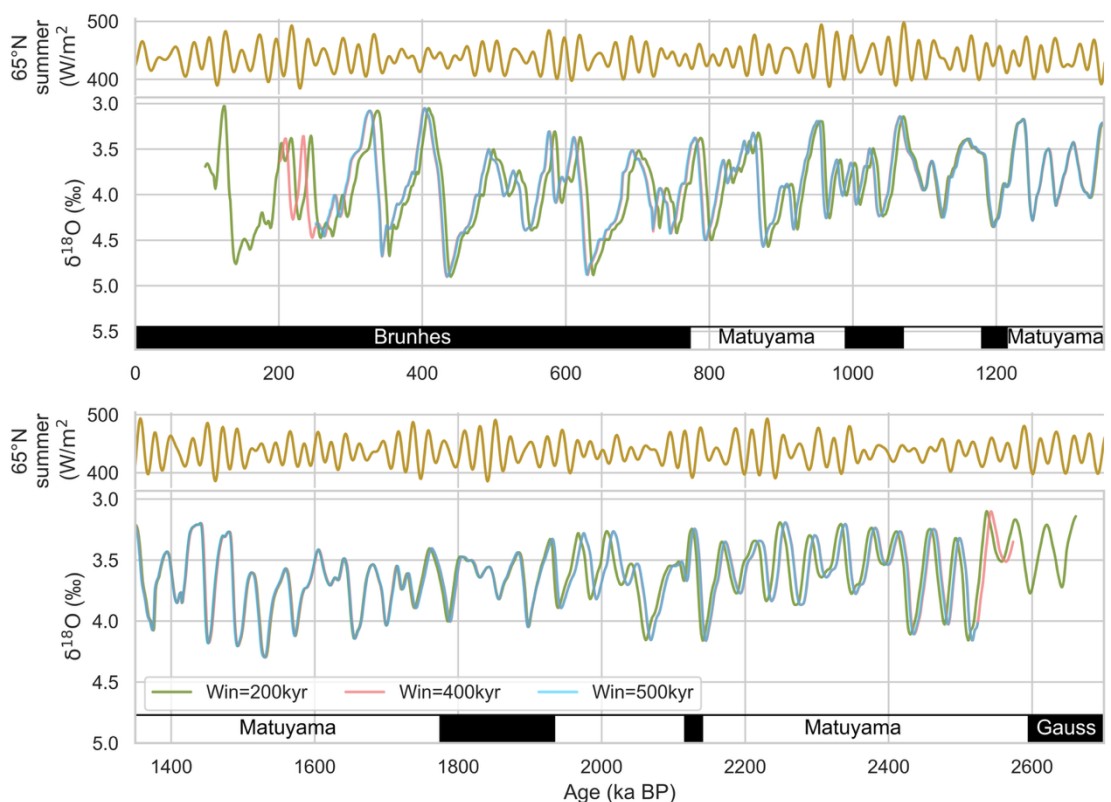

**Figure 4. Comparison of the effect of different window sizes in using eTimeOpt. The blue, pink, and olive lines are the auto-tuned stacks with different window sizes. The pink and blue lines mostly overlap each other. The yellow line is Northern Hemisphere insolation at 65°N.**

The geomagnetic reversal and excursion ages rely on published data using astronomical tuning (Table 1), as reported in Ogg (2020). These ages are mostly consistent with the ages produced by the radioisotopic Ar/Ar dating (Singer, 2014; Channell et al., 2020).

The untuning process starts with BIGSTACK$_{LR04}$. To apply the magnetic reversal ages to BIGSTACK$_{LR04}$, the age estimates of 35 of BIGSTACK$_{LR04}$'s constituent records with high-resolution paleomagnetic measurements are first linearly interpolated according to the geomagnetic reversal ages at 26-kyr intervals. These 35 cores were selected based on the availability of high-resolution paleomagnetic measurements and are all from either the Atlantic or Pacific Ocean. Linear interpolation between



magnetic reversals is performed using an adjusted depth scale that has been corrected for down-core sediment compaction as described above. The averages of the interpolated ages form an initial untuned age model applied to BIGSTACK$_{LR04}$.

The age uncertainty of the magnetic reversal age model is relatively large between bracketing geomagnetic reversals and was estimated to be ~9 kyr in the middle of the Brunhes chron for the untuned stack of Lisiecki (2010). Therefore, we checked the quality of this portion of our BIGMACS$_{magrev}$ age model by comparing it with the H23NA benthic $\delta^{18}$O stack, which is aligned

to the radioisotopically-dated speleothem records from 0-654 ka (Hobart et al., 2023) and not astronomically tuned. Based on this comparison, we added an age model tie point for Termination IV which shifts its age 12 kyr younger in our BIGMACS$_{magrev}$ stack, to better agree with H23NA. Note that H23NA is based on correlation to dated speleothems, and thus is independent of astronomical forcing assumptions. Finally, the resulting BIGMACS$_{magrev}$ age model produces variable-length time steps between the age-shifted stack data with an average 1-kyr spacing and a standard deviation of 0.07 kyr. We apply piecewise

linear interpolation to the BIGSTACK$_{LR04}$ $\delta^{18}$O values, yielding an even 1-kyr spacing for the BIGMACS$_{magrev}$ age model. We name this untuned stack BIGSTACK$_{magrev}$.

### 3.3 Auto-tuned δ18O stack

Next, the BIGMACS$_{magrev}$ stack is "minimally tuned" to the pervasive ~41 kyr obliquity cycle in the $\delta^{18}$O stack, using the automated eTimeOpt statistical algorithm of the software Astrochron (Meyers, 2015, 2019). The algorithm's name is short for

"evolutive time scale optimization," and here it is used to evaluate the spectral power of ~41 kyr obliquity forcing in BIGMACS$_{magrev}$. Note that the input data for eTimeOpt are normally on a depth scale; here, we substitute age of the BIGMACS$_{magrev}$ stack for depth because the BIGMACS$_{magrev}$ ages should scale linearly to the compaction-corrected mean depth of the stacked cores. We choose to tune to obliquity, and not eccentricity or precession, because the BIGMACS$_{magrev}$ stack shows persistent power around 41 kyrs throughout the Pleistocene (Fig. 3). The eTimeOpt algorithm requires the input

of a window size for its evolutive analysis of secular changes in the sedimentation rate throughout the stratigraphy. We tested several window sizes and found 400 kyrs to be the most appropriate (Fig. 4). Importantly, this approach corrects for secular shifts in sedimentation rate on the scale of 400 kyr, but unlike the manually tuned stack (Section 3.4), does not modify phase relationships that are inherent to the $\delta^{18}$O stack or impose phase responses to the forcing. The resulting auto-tuned stack has a resolution of about 1 kyr and is interpolated to 1-kyr regular time steps (via piecewise linear interpolation).


Because eTimeOpt only produces a floating timescale by stretching or compressing the stack to concentrate obliquity power, the timescale is not anchored to any absolute ages. We anchor the auto-tuned stack by minimizing the root mean square error (RMSE) fit to the geomagnetic ages (Table 1). The best fit to the geomagnetic ages is determined by sliding the eTimeOpt timescale in steps of 0.1 kyr over a range of -10 kyr to 10 kyr and finding the age shift that minimizes the RMSE. Because the

400-kyr tuning window causes truncations at the top and bottom 200 kyrs of the auto-tuned stack, we spliced the top and





| Age | BIGSTACK$_{mixed}$ tuning targets | BIGSTACK$_{mixedA}$ alignment targets | BIGSTACK$_{mixedP}$ alignment targets |
|---|---|---|---|
| 0-150 ka | LS16 global stack | LS16 deep North Atlantic stack | LS16 deep Pacific stack |
| 150-654 ka | H23NA | H23NA | H23NA lagged by 1 kyr |
| 654-2700 ka | Imbrie and Imbrie (1980) ice model with Zeebe and Lourens (2022a, b) insolation values | BIGSTACK$_{mixed}$ with Atlantic splice (Zhou et al., 2024) for 1.8-1.9 Ma | BIGSTACK$_{mixed}$ with Pacific splice (Zhou et al., 2024) for 1.8-1.9 Ma |

**Table 2. Alignment and tuning targets for manually tuned stacks. LS16 refers to benthic δ$^{18}$O stacks from (Lisiecki and Stern, 2016) and H23NA refers to a North Atlantic benthic δ$^{18}$O stack from Hobart et al (2023). Both studies produced age models without astronomical tuning assumptions based on alignment to millennial-scale variability in speleothems. The Atlantic and Pacific stacks were constructed using existing stacks as initial alignment targets, with the 1.8-1.9 Ma portion of BIGSTACK$_{mixed}$ replaced with the respective regional stacks from Zhou et al (2024).**

bottom 200 kyrs of the global BIGSTACK$_{mixed}$ stack (next section) into the auto-tuned stack. We name this stack BIGSTACK$_{auto}$.

To assess the age uncertainty associated with the automated tuning used for BIGSTACK$_{auto}$, we conducted Monte Carlo simulations on the eTimeOpt-derived age model using different plausible durations of the obliquity cycle based on an existing model (Waltham, 2015). At 1388.5 ka, the dominant obliquity cycle has a mean value of 40.95 kyr and a standard deviation of 0.1 kyr. From a Gaussian distribution with this mean and standard deviation, we drew 1000 sample cycle lengths as input to eTimpOpt. The floating time scales generated by eTimeOpt were then anchored using the same procedure minimizing the RMSE fit to geomagnetic ages.

### 3.4 Manually aligned δ$^{18}$O stack with mixed age constraints

While the auto-tuned stack focuses on concentrating spectral power attributed to obliquity forcing, an alternative strategy is to manually tune the stack using the best available age constraints throughout the Pleistocene. Manual tuning allows for differentiating tuning targets by time periods and geographical locations and for more fine-scale age control (e.g., alignment of individual glacial cycles).

We divide the Pleistocene into three segments and set tuning targets most suitable for each (Table 2). The targets for the youngest 654 kyrs (Lisiecki and Stern, 2016; Hobart et al., 2023) are chosen because they are supported by alignments to radioisotopically dated speleothem records. Additionally, the LS16 δ$^{18}$O global stack is volume-weighted, thus avoiding the potential bias of oversampling the Atlantic compared to the Pacific. The LS16 stacks have a mean 2σ uncertainty half-width



of 2.6 kyr for the North Atlantic and 3.6 kyr for the global and deep Pacific stacks. The H23NA $\delta^{18}$O stack is a regional North Atlantic stack with an average $2\sigma$ uncertainty half-width of 3.6 kyr during deglaciations and ~6 kyr in between.

Because no absolutely dated targets for benthic $\delta^{18}$O are available from 2700 ka to 654 ka, we tune to a target produced using
the ice model of Imbrie and Imbrie (1980), which was also used to construct the astronomical tuning target for the LR04 stack,

$$\partial y \, / \, \partial t = (1 \, +/- \, b)/T * (x\text{-}y)$$

where y is the ice volume, x is the insolation, t is time (in kyr), T is the ice response time (in kyr), and b is a nonlinearity coefficient. However, unlike the ice model used by the LR04 stack, we use the insolation values produced by newer studies, published with open-sourced data and code (Zeebe and Lourens, 2022a, b). Furthermore, we change the model parameter value
for T to be 4 kyr and b to be 0.5, values similar to those recommended by Lisiecki and Stern (2016). Additionally, because the target period is mostly before and during the MPT, we incorporate the ice volume assumptions of the "antiphase hypothesis" (Raymo et al., 2006), another departure from the methodology employed by the LR04 stack. The hypothesis, which has found support in subsequent studies that detected precession signals in the early Pleistocene climate (Martínez-Garcia et al., 2010; Patterson et al., 2014; Shakun et al., 2016; Zhou et al., 2024), calls for an Antarctic ice sheet forced by local summer insolation
that varied in ice volume by about the equivalent of 30 m of sea level change. We thus mix the Northern and Southern Hemisphere (NH and SH) model results by a ratio of 8:3, assuming an 80 m sea-level-equivalent variation on NH ice sheet size (Supplemental Data File 3). Note that only the ratio of the NH and SH ice volumes is relevant to the ice model, whereas the absolute values are not relevant because the ice model output is not scaled directly to $\delta^{18}$O values.

For the hand-tuned global stack, we use QAnalySeries (Pälike and Kotov, 2024) to align the BIGMACS$_{magrev}$ stack to the three different targets. Care was taken to ensure that age differences among the targets would not create conflicts during the alignment. Visual examination reveals that the concatenation between the LS16 and H23NA stacks at 136 ka is in good agreement, but the junction between the age models using the H23NA stack and the ice model at 654 ka shows discrepancy. Either the age model tuned to the H23NA stack is too young or the age model tuned to the ice model is too old; this discrepancy
leads to a 3-kyr-long discontinuity. Our approach to tackle the issue is to leave a 5-kyr gap at the concatenation junction, which we fill with 'NaN' (not a number). While this leaves out a small amount of data in our stack, our approach avoids creating





artificial features at the concatenation point. Because this age model incorporates multiple types of age

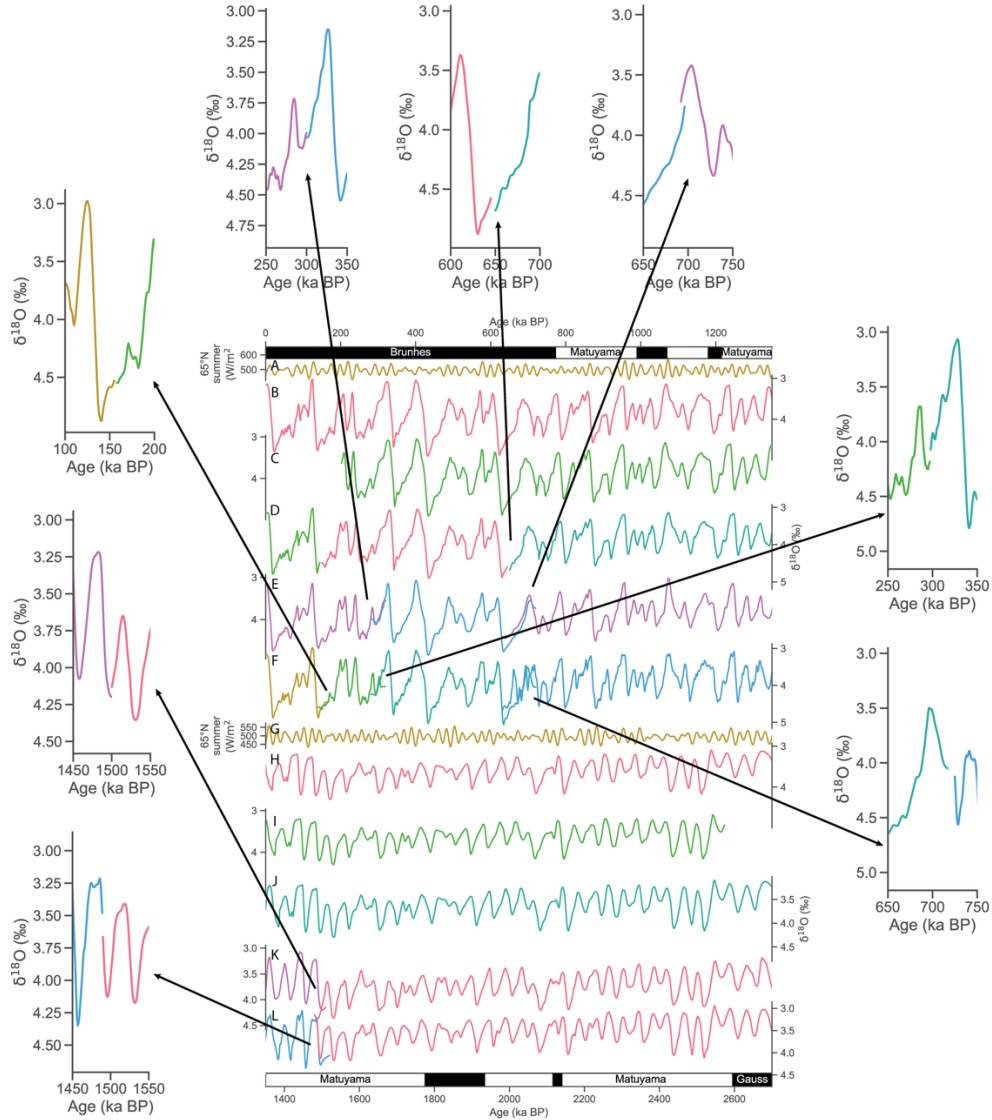

**Figure 5. Five new Pleistocene benthic δ¹⁸O stacks showing where segments of stacks are joined. (A) Northern Hemisphere insolation at 65°N. (B) BIGSTACK_magrev. (C) BIGSTACK_auto. (D) BIGSTACK_mixed. (E) BIGSTACK_mixedP. (F) BIGSTACK_mixedA. (G-L) Same as above but from 1350 to 2700 ka. Geomagnetic polarity chrons are marked at the top and bottom of the figure, with the four subchrons within Matuyama being, from young to old, Jaramillo, Cobb Mountain, Olduvai, and Réunion (Feni). The inset panels show how the stack segments are trimmed and concatenated. When there are overlapping stack segments after trimming, the overlapped portions are averaged during concatenation.**

constraints (speleothem-based ages and an astronomically forced ice model), we name the global stack created this way BIGSTACK_mixed.





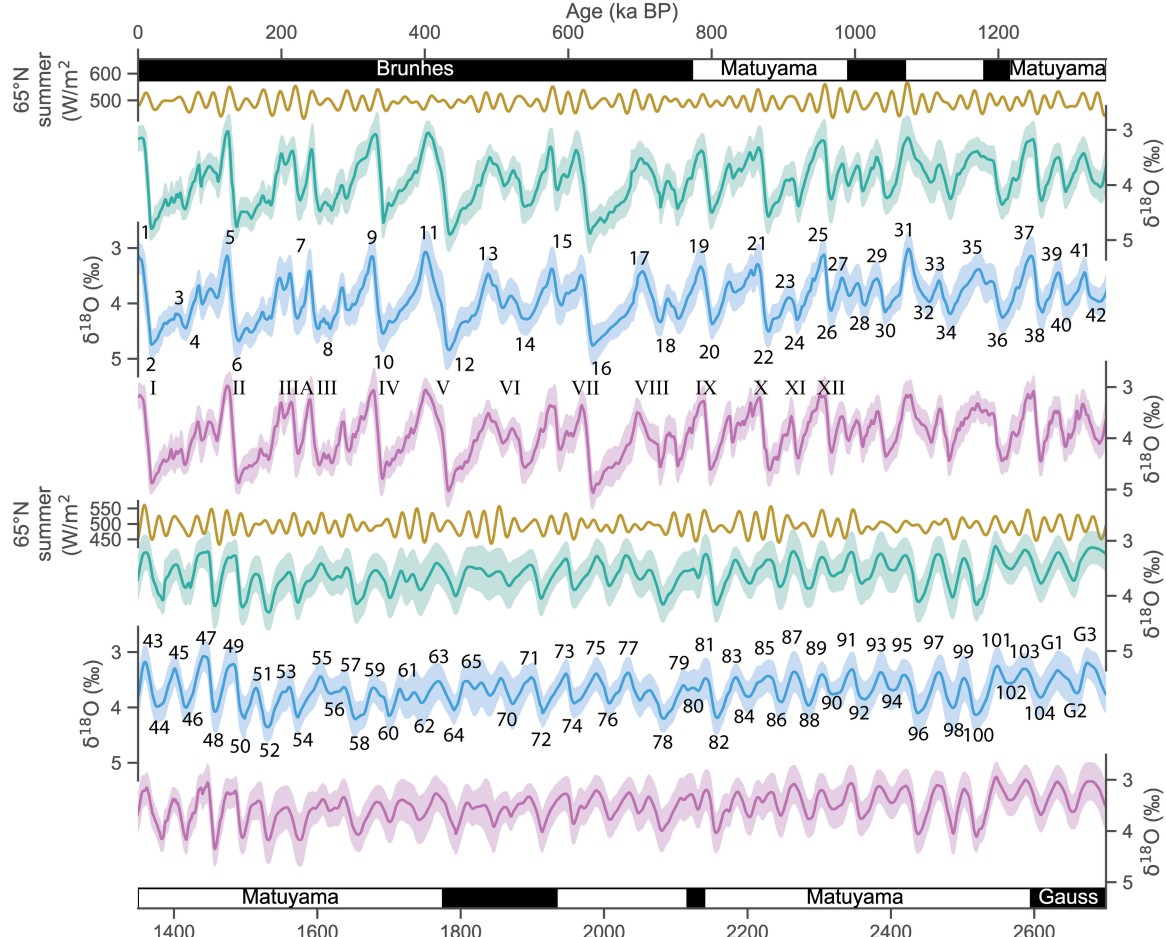

**Figure 6. BIGSTACK_{mixed} (green), BIGSTACK_{mixedP} (blue), and BIGSTACK_{mixedA} (purple) and their 95% credible intervals (shade). Note that the shown uncertainty of BIGSTACK_{mixed} is inherited from BIGSTACK_{LR04} and does not include the uncertainty of the untuning and hand-tuning processes.**

BIGSTACK_{mixed} is also used as the initial alignment target for the Atlantic and Pacific stacks from 655-2700 ka, except for regional splices from 1.8-1.9 Ma. The BIGSTACK_{mixedA} and BIGSTACK_{mixedP} stacks likewise have gaps over some junctions between alignment targets (Fig. 5). The gaps range in length from 3 kyrs at around 156 ka in BIGSTACK_{mixedA} to 8 kyrs at around 721 ka in BIGSTACK_{mixedA}.





**Figure 7. Five new Pleistocene benthic δ¹⁸O stacks. (A) Northern Hemisphere insolation at 65°N. (B) BIGSTACK_magrev. (C) BIGSTACK_auto. (D) BIGSTACK_mixed. (E) BIGSTACK_mixedP. (F) BIGSTACK_mixedA. (G-L) Same as above but from 1350 to 2700 ka. Geomagnetic polarity chrons are marked at the top and bottom of the figure, with the four subchrons within Matuyama being, from young to old, Jaramillo, Cobb Mountain, Olduvai, and Réunion (Feni). The marine isotope stages are marked by numbers, and the Roman numerals are the glacial terminations. The arrows point out noticeable differences between stacks. The gray line in each panel is the global LR04 stack for comparison.**





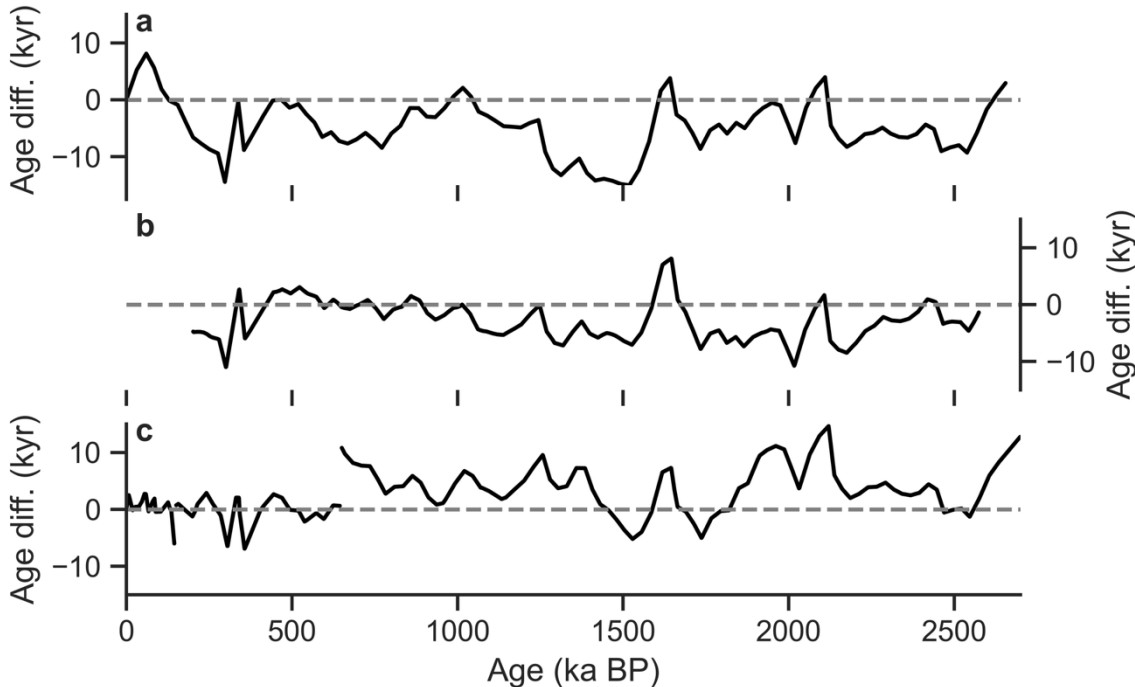

**Figure 8. Age differences of BIGSTACK$_{magrev}$, BIGSTACK$_{auto}$, and BIGSTACK$_{mixed}$ relative to BIGSTACK$_{LR04}$. (a) The**
**BIGSTACK$_{magrev}$ age minus BIGSTACK$_{LR04}$ age on the BIGSTACK$_{untuned}$ time scale. (b) The BIGSTACK$_{auto}$ age minus**
**BIGSTACK$_{LR04}$ age on the BIGSTACK$_{auto}$ time scale. (c) The BIGSTACK$_{mixed}$ age minus BIGSTACK$_{LR04}$ age on the**
**BIGSTACK$_{mixed}$ time scale. The curve breakpoint is where the three BIGSTACK$_{mixed}$ segments are connected.**

## 4 Results

The uncertainty of BIGSTACK$_{mixed}$, BIGSTACK$_{mixedA}$, BIGSTACK$_{mixedP}$ represents the 95% confidence interval for $\delta^{18}O$
values generated from Markov Chain Monte Carlo (MCMC) samples (Fig. 6). For BIGSTACK$_{mixed}$, the one-sided 95%
confidence interval typically ranges from 0.21 ‰ to 0.54 ‰ and almost always increases with age. These uncertainties are
transferred from the generation of BIGSTACK$_{LR04}$ with age shifts derived from the hand-tuned age model adjustments and do
not include uncertainties associated with the tuning procedure. Because BIGSTACK$_{mixedA}$ and BIGSTACK$_{mixedP}$ were
generated using the BIGSTACK$_{mixed}$ age model target, no adjustments were needed to the confidence intervals generated by
BIGMACS.

BIGSTACK$_{LR04}$, the stack that serves as the basis for BIGMACS$_{magrev}$, deviates substantially from the LR04 age model in just
one place, shifting 5-9 kyr older at ~1.1 Ma (MIS 33; Fig. S1). However, the five versions of the Pleistocene benthic $\delta^{18}O$





stacks with updated age models show differences in the timing and shape of multiple marine isotope stages (Fig. 7).

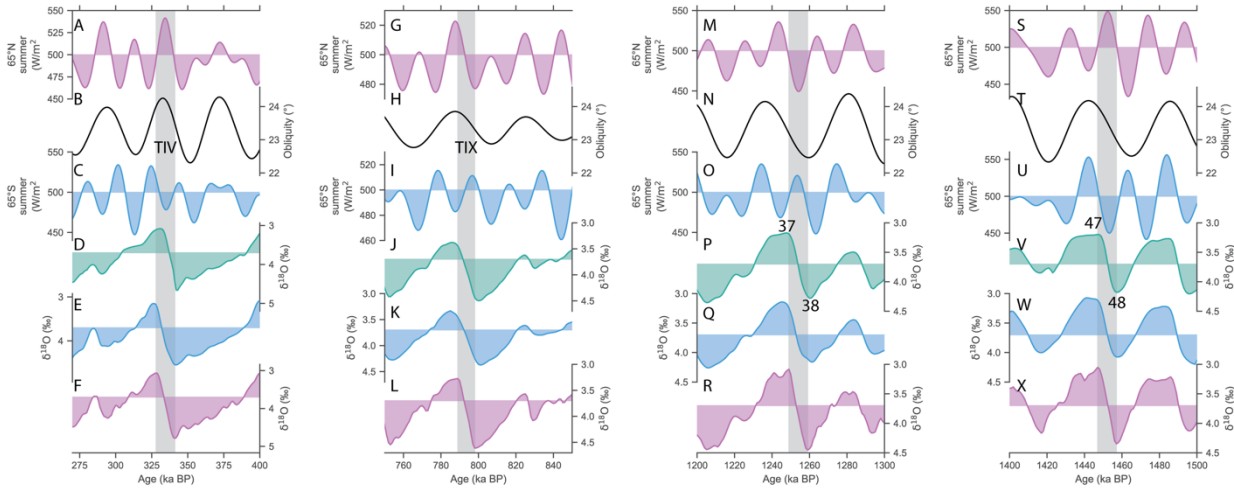


**Figure 9. Instances of the deglaciation in BIGSTACK$_{mixedA}$ being shorter than BIGSTACK$_{mixedP}$. (A) 65°N summer insolation. (B) Obliquity. (C) 65°S summer insolation. (D) BIGSTACK$_{mixed}$. (E) BIGSTACK$_{mixedP}$. (F) BIGSTACK$_{mixedA}$. (G-L) Same records for Termination IX. (M-R) Same records for the termination from MIS 38 to 37. (S-X) Same records for the termination from MIS 48 to 47.**

Since BIGSTACK$_{auto}$ and BIGSTACK$_{mixed}$ only update the timing of the BIGMACS$_{magrev}$ global stack, the three stacks do not differ in shape, except for the minor changes caused by the interpolation onto regular time steps. The Atlantic and Pacific stacks have different input records and are thus different in both timing and shape.

Timing-wise, the BIGSTACK$_{magrev}$ stack is shifted younger than BIGSTACK$_{LR04}$ by less than 15 kyr during 140-2650 ka,
except at ~1000 ka, ~1640 ka, and ~2100 ka, when the BIGSTACK$_{LR04}$ ages are younger by several kiloyears (Fig. 8). The age models for BIGSTACK$_{LR04}$ and the auto-tuned stack generally agree to within 10 kyr (Fig. 8). The timing difference between BIGSTACK$_{LR04}$ and the auto-tuned stack reaches maxima of 9 kyr during MIS 57 (BIGSTACK$_{LR04}$: 1633 ka vs. BIGSTACK$_{auto}$: 1642 ka) and 12 kyr during MIS 76 (BIGSTACK$_{LR04}$: 2005 ka vs. auto-tuned: 1993 ka).

BIGSTACK$_{mixed}$ is shifted older than BIGSTACK$_{LR04}$ by 10 kyr or more across only three brief intervals – during 1913-1978 ka, 2056-2119 ka, and 2653-2700 ka (Fig. 8). BIGSTACK$_{mixed}$ is younger than BIGSTACK$_{LR04}$ during 260-400 ka. BIGSTACK$_{mixed}$ is generally older than BIGSTACK$_{LR04}$ during 600-2700 ka (except for 1439-15881 ka and 1675-1815 ka), with a maximum offset of 15 kyr at MIS 79.  Because the 0-100 ka and 2600-2700 ka segments of BIGSTACK$_{auto}$ are spliced from the hand-tuned stack, the two stacks are identical during these intervals.






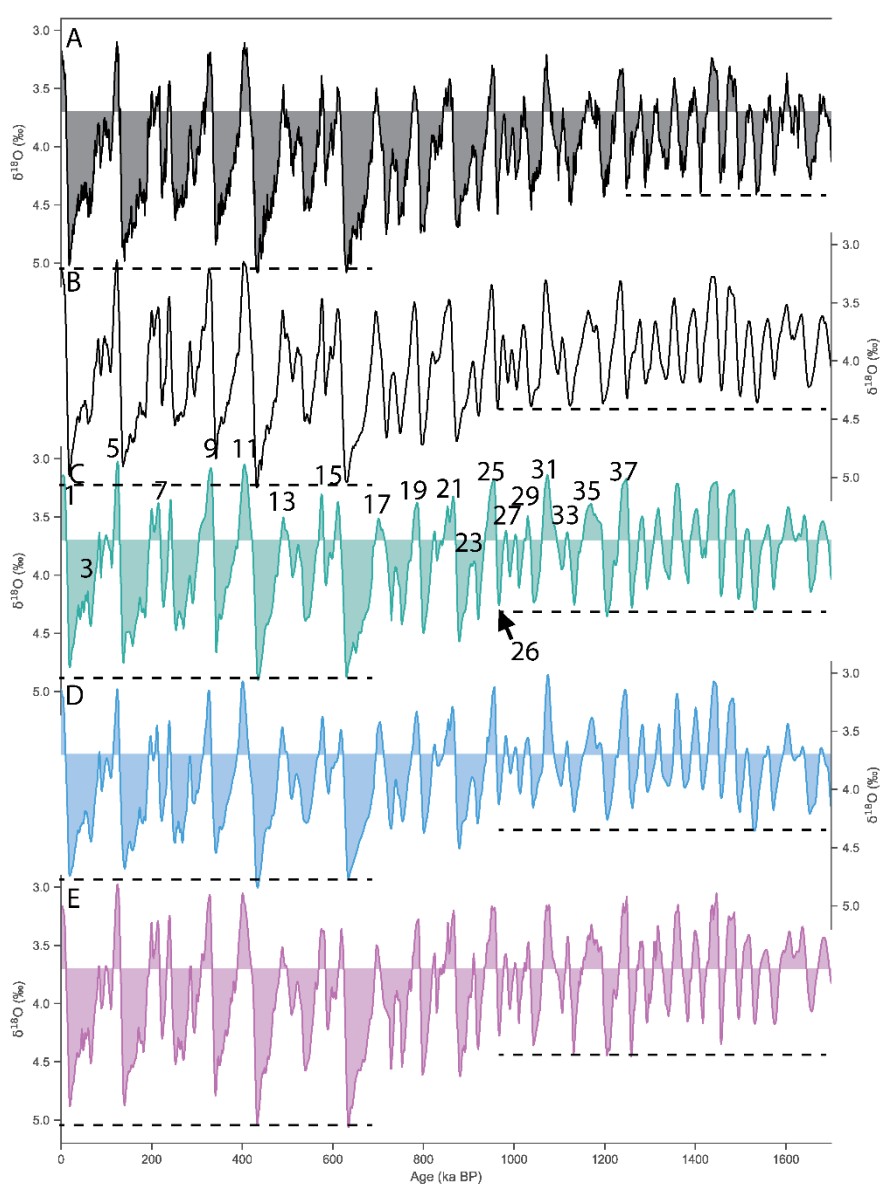

**Figure 10. Comparison of LR04, a smoothed version of LR04, and our new stacks during the MPT. (A) LR04. (B) LR04 applied with a Savitzky-Golay filter of 10-kyr window size and polynomial order of 3. (C) BIGSTACK$_{mixed}$. (D) BIGSTACK$_{mixedP}$. (E) BIGSTACK$_{mixedA}$. The marine isotope stages are marked by numbers. The horizontal dashed lines mark the typical glacial δ$^{18}$O values during the 41-kyr world and late Pleistocene in each stack.**




Because BIGSTACK$_{mixed}$, BIGSTACK$_{mixedP}$, and BIGSTACK$_{mixedA}$ share targets that have the same or very similar timings, the age model difference between the three stacks is small compared to BIGSTACK$_{magrev}$ and BIGSTACK$_{auto}$. On the other hand, the shapes of the three hand-tuned stacks differ subtly because they include different sets of cores. During MIS 3, the Pacific stack shows less variability than the global and Atlantic stacks. The global and Atlantic stacks show similar $\delta^{18}O$ values

between MIS 5a and 5c, but in the Pacific stack, MIS 5c appears more enriched in $\delta^{18}O$ than MIS 5a. Additionally, the global and Atlantic stacks display similar $\delta^{18}O$ values among MIS 7a, 7c, and 7e, but in the Pacific stack, MIS 7a is more enriched in $\delta^{18}O$ than MIS 7c and 7e. For a detailed designation of the lettered marine isotope substages, see Railsback et al. (2015). The duration of Termination IV is 3 kyrs shorter in the Atlantic stack (341-328 ka from MIS 10 $\delta^{18}O$ maxima to MIS 9 $\delta^{18}O$ minima) than in the Pacific stack (342-326 ka) (Fig. 9). This is also the case for Termination IX, which is shorter in the Atlantic

stack (789-798 ka) than the Pacific stack (785-801 ka). While MIS 13b shows more enriched $\delta^{18}O$ values in the Pacific stack than the Atlantic stack, the trend is reversed during MIS 14. Lastly, while MIS 15a and 15e are similar in $\delta^{18}O$ values in the Atlantic stack, MIS 15a is more depleted in $\delta^{18}O$ compared to 15e in the Pacific stack. Beyond the MPT, the terminations from MIS 38 to 37 and from MIS 48 to 47 are also shorter in duration in the Atlantic stack than the Pacific stack.

The new stacks also differ from the LR04 stack during some glacial cycles (Fig. 7) in addition to the regional differences at 1.8 Ma described by Zhou et al. (2024). Some of the glacials of our new stacks are smaller in magnitude in the early Pleistocene, notably during MISs 46, 56, 62, 65, and 70. Additionally, the progression of increasing glacial magnitude during the Mid-Pleistocene Transition (MPT) appears more abrupt in our new stacks compared to the LR04 stack (Fig. 10). At the younger end of the MPT, both the LR04 and our stacks agree with MIS 16 being the first large-amplitude glacial stage of the 100-kyr

world. However, MIS 38 is the last glacial stage in the LR04 stack whose amplitude is similar to those of the 41-kyr world. In contrast, glacials with 41-kyr world amplitudes persisted as late as MIS 26 in the new stacks. This discrepancy during the MPT compared to the LR04 stack appears to be an artifact of smoothing that occurs during the BIGMACS stack construction rather than a contradictory signal against LR04. This is apparent when LR04 is smoothed by various methods and compared to the hand-tuned global stack (Fig. 10).

**5 Discussion**

**5.1 Age Model Uncertainty**

Our newly constructed stacks must be interpreted in the context of the assumptions and uncertainties associated with the methods that produced them. The BIGSTACK$_{LR04}$ that serves as the starting point of our stacks is probabilistic and includes uncertainty associated with the alignment process but not age model construction.


Several sources of uncertainty affect age estimates of BIGSTACK$_{magrev}$, which is constructed with 19 magnetic reversals unevenly spaced across 2.7 Ma. All of the geomagnetic reversal/excursion ages we use were determined by astronomical





tuning (Cande and Kent, 1995; Ogg, 2020). This means that our BIGSTACK$_{magrev}$ stack utilizes a small number of astronomically tuned age-control points derived from prior studies. We attempted to construct an age model with only Ar/Ar-based geomagnetic reversal ages, but the resulting stack created tuning difficulties for the eTimeOpt algorithm in the interval 1.9-2.1 Ma.

Identifications of magnetic reversal depths in individual cores also have some uncertainty, but most reported identifications do not come with uncertainty estimates, which precludes the quantification of the associated age uncertainty for the BIGSTACK$_{magrev}$ stack. Additionally, by adding a tie point at Termination IV according to the age model of H23NA, we assume that the termination timing of the deep North Atlantic stack is broadly representative of the global ocean. Although the timing of the last deglaciation differs regionally by up to 4 kyr (Lund et al., 2015; Rand et al., 2024), this tie point likely reduces the age uncertainty of BIGSTACK$_{magrev}$ without introducing any additional astronomically-derived age information.

Ages between magnetic reversals are estimated based on the assumption of constant mean, compaction-corrected sedimentation rates. If sedimentation rates at individual sites are largely independent of one another and/or if a relatively fixed total sediment supply is spatially redistributed among sites, we expect the average across a large enough pool of sites to produce a global mean sedimentation rate that is approximately constant throughout the entire stack; this is an important constraint on the age model. Although our BIGSTACK$_{magrev}$ compilation consists of 35 core records, there is no guarantee that our sampling is sufficient to fully satisfy the constant global mean sedimentation rate constraint. The sites available for constraining ages in the BIGSTACK$_{magrev}$ stack are also affected by selection bias. While pelagic limestones and calcareous/siliceous oozes are efficient recorders of the paleo-geomagnetic field, Pacific red clay lacks remanence-carrying minerals (Opdyke and Channell, 1996). This difference in the quality of the paleomagnetic measurements from different sedimentation environments introduces selection biases in the compilation, which can skew the stochasticity assumption that would produce constant mean sedimentation rate overall. Additionally, our formulation of the compaction correction is very much a simplified version of the real-world process, as the availability of porosity data is limited, and the existing data show complex porosity-depth relations (Huybers and Wunsch, 2004).

The age uncertainty associated with the automated tuning used for BIGSTACK$_{auto}$ after anchoring of the stack ranges from 0.2 to 3 kyr (Fig. 11) in Monte Carlo simulations. The standard deviation of the time uncertainty is lowest at 1400 ka, likely because it is the mid-point of the geomagnetic ages used to anchor the floating time scale. The time uncertainty increases towards both the younger and older ends of the stack, with a standard deviation of 3 kyr at the youngest end and 2.6 kyr at the





oldest end. The 400-kyr moving window used to identify and concentrate obliquity power does not allow explicit

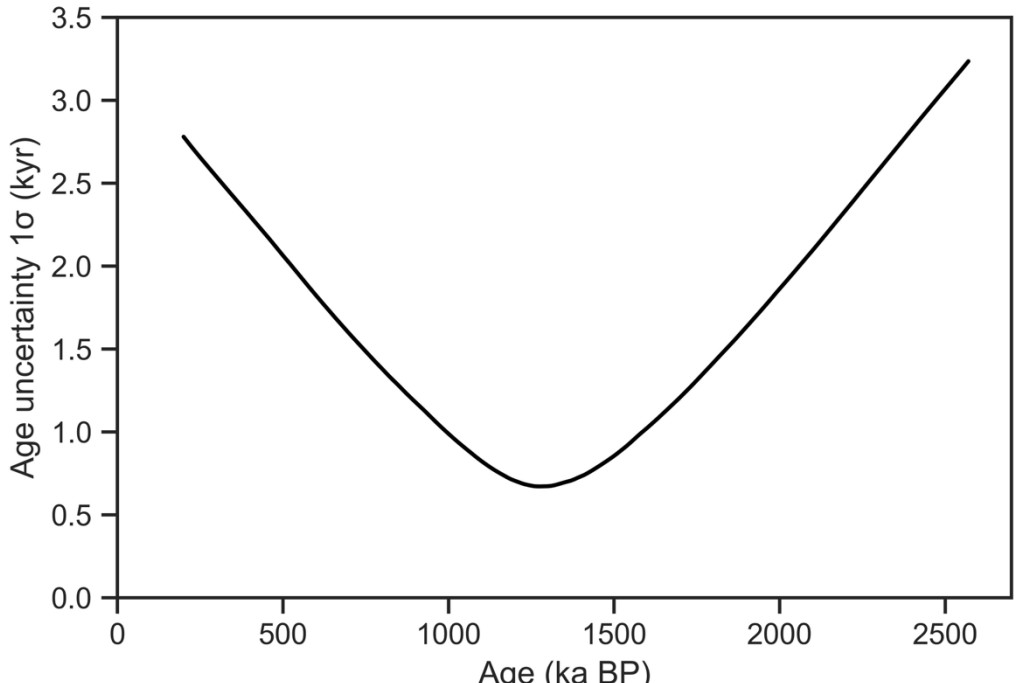


**Figure 11. Age uncertainty (1σ) of the eTimeOpt auto-tuned stack anchored to the paleomagnetic reversal data. Calculations use a Monte Carlo procedure that considers the 41-kyr obliquity period tuning uncertainty, signal/noise, and resolution limitations of the data. These should be considered 400-kyr-secular estimates of timescale uncertainty.**

estimation of age uncertainty associated with sedimentation rate variability on timescales shorter than the moving window;

thus it provides secular estimates.

Rigorous age uncertainty estimates cannot be constructed for BIGSTACK$_{mixed}$ due to the manual nature of its construction. Our manual tuning process uses as few tie points as possible to avoid creating abrupt changes in the sedimentation rates, with tie points primarily placed on features such as high-amplitude glacials and interglacials. However, the tie points used to create

the target stacks for the youngest segment of the stack between 0 and 654 ka (Lisiecki and Stern, 2016; Hobart et al., 2023) coincide with Heinrich events, which sometimes occur during relatively stable benthic $\delta^{18}$O intervals. This mismatch in the types of features used for tie points between the target stack and tuning means uncertainty in the hand-tuned age model will be higher than if the targets and tuning use the same set of tie points. For example, when the North Atlantic ice-rafted debris (IRD) events are aligned to Asian weak monsoon intervals (Lisiecki and Stern, 2016; Hobart et al., 2023), these events during

a glacial period may have relatively lower uncertainty compared to an interglacial that is used as a tie point during the tuning process. However, the IRD events may not be associated with apparent features in the benthic $\delta^{18}$O record suitable for creating



tie points. In hand tuning our stacks to the H23NA stack and the LS16 global stack, we also rely on their assumptions that the correlation of climatic events from absolutely dated archives to marine sediments is relatively well constrained and that phase lags within the climate system are small.


We lagged the H23NA North Atlantic stack by 1 kyr as the target for BIGSTACK$_{mixedP}$ (Table 2), but the magnitude of benthic $\delta^{18}O$ lag between the Atlantic and Pacific is uncertain. Previous studies have estimated the lag during Termination 1 to be as high as 3.9 kyr (Skinner and Shackleton, 2005), as low as 1 kyr (Stern and Lisiecki, 2014), or somewhere in between (Rand, 2023). These estimates focus on the last deglaciation because radiocarbon provides absolute age models. A tracer transport

model demonstrated that regional surface boundary conditions can reproduce the high end of the lag estimates (Gebbie, 2012). However, the benthic $\delta^{18}O$ lag between the ocean basins may have been at its maximum during terminations (Stern and Lisiecki, 2014), judging from the regional stacks of the last glacial cycle. We thus choose a more conservative 1 kyr as a constant mean Pacific-Atlantic lag. We note that the uncertainty regarding the lag can be reduced if there is a Pacific stack aligned to absolutely dated records that spans multiple glacial cycles of the late Pleistocene. Siku events, proposed as precursors

to Heinrich events, have so far only been reported during the last glacial (Walczak et al., 2020) but may have the potential to quantify the Atlantic-Pacific lag during previous glacial periods. The alignment of sea surface temperature to Antarctic ice core records could also prove valuable if synchronous changes between the two can be demonstrated.

In the Imbrie and Imbrie (1980) ice model used as the tuning target for BIGSTACK$_{mixed}$ from 654-2700 ka, our choice of 4

kyr for the ice response time is much shorter than the 15-kyr response time utilized by the LR04 during 0-1.5 Ma and a ramping response time from 5 to 15 kyrs during 1.5-3 Ma. The shorter response time was proposed by comparing the ice model during 0-150 ka with a global stack aligned to absolutely dated records (Lisiecki and Stern, 2016). It is possible that ice response time might be shorter during the early Pleistocene because the ice sheets were smaller, thus making our target's age younger than reality. Nevertheless, the ice response time cannot decrease much further without risking an unrealistic scenario where the ice

response behavior becomes nearly instantaneous. A different choice of nonlinearity parameters would also impact the best-fit estimate of response time.

Our regional stacks use BIGSTACK$_{mixed}$ as the target during 654-2700 ka except 1.8-1.9 Ma. The Atlantic and Pacific benthic $\delta^{18}O$ records throughout the Pleistocene have been observed to largely covary except between 1.8-1.9 Ma and at ~2.05 Ma

(Zhou et al., 2024). A comparison between a benthic $\delta^{18}O$ stack of five Ceara Rise cores and the LR04 global stack found that the only periods of notable differences between the two are during 1.8-1.9 Ma and 4.0-4.5 Ma (Wilkens et al., 2017). These observations support the use of the same target for our regional hand-tuned stack except for the interval 1.8-1.9 Ma.

During 1.8-1.9 Ma, the regional stacks use data spliced from previously constructed Atlantic and Pacific stacks that used the

same cores as this study (Zhou et al., 2024). The spliced portions of the stack use the previously published age model with tie



points to obliquity for MIS 64 and 74 and NH insolation minima for MIS 68 and 70 (Atlantic stack only), rather than the ice model used for alignment of BIGSTACK$_{mixed}$. While the exact timing of the glacial cycles and the response time from astronomical forcing in these regional stacks are uncertain, Zhou et al (2024) selected these tie points to minimize average (normalized) sedimentation rate changes in the regional stacks. Based on detailed stratigraphic analysis Zhou et al. (2024)

concluded that differences in the glacial amplitudes of these regional $\delta^{18}O$ stacks are unlikely to be a result of age model uncertainty.

The shorter durations of some deglaciations in BIGSTACK$_{mixedA}$ compared to BIGSTACK$_{mixedP}$ could potentially be explained two ways (Fig. 9). As ice sheets retreat, the lighter $\delta^{18}O$ may enter the deep Atlantic faster than the deep Pacific due to the

differences in ventilation rates (Skinner and Shackleton, 2005). While this can explain the shorter duration of the deglaciations in the Atlantic stack, it cannot explain the seemingly later start of Atlantic deglaciations. However, the targets of BIGSTACK$_{mixedA}$ and BIGSTACK$_{mixedP}$ are not constructed from speleothem-based absolute dates beyond 654 ka, and a simplistic constant temporal offset of 1 kyr is stipulated between the targets during 150-654 ka. The later start of the Atlantic deglaciations may thus be an artifact of the stack construction process. Another reason for the shorter duration of the

BIGSTACK$_{mixedA}$ deglaciation could be the disparate changes in the polar source water properties (Gebbie, 2012). In other words, regional changes in seawater $\delta^{18}O$ and temperature may not be synchronous between the North Atlantic and Southern Oceans. While the subsurface Pacific is primarily ventilated by water sourced from the Southern Ocean, the mid-deep Atlantic water mass has a northern origin.

Regarding the age differences between the new stacks and LR04 (Fig. 8), a shift toward somewhat older ages in the new stacks is expected based on the smaller time constant used in the updated ice volume model. Another likely cause for age differences compared to LR04 is the conservative tuning strategy used for LR04 that minimized global sedimentation changes instead of strictly matching the timing of changes in benthic $\delta^{18}O$ and the ice volume model. New insolation solutions and the incorporation of anti-phased precession effects in the ice volume model may also contribute to some subtle age differences.

**5.2 Applications**

Our five versions of the Pleistocene benthic $\delta^{18}O$ stacks provide a new framework for a wide range of applications in paleoceanographic hypothesis testing. Furthermore, comparing the three age models for the global stack clarifies the time shifts associated with different aspects of age model development. BIGSTACK$_{magrev}$ is useful for assessing the climatic response to astronomical forcing with less risk of circular reasoning. Although the geomagnetic reversal/excursion ages have

been astronomically tuned, this is unlikely to artificially introduce significant astronomical power or phase coherence to the stack during periods when the presence of geomagnetic reversals/excursions are sparse. During 773-1070 ka and 1775-1934 ka, multiple geomagnetic reversals and excursions occurred in a relatively short period of time, and caution should be taken



when using these portions of the stack to assess the climatic response to astronomical forcing. The TIV tie point from Hobart et al (2023) does not add any astronomically derived age information.


BIGSTACK$_{auto}$ concentrates the ~41 kyr obliquity power in the $\delta^{18}$O stack with minimal assumptions or subjective intervention and does not impose phase responses to the forcing. Because BIGSTACK$_{auto}$ makes no adjustments related to orbital eccentricity or precession, its age model may be useful for analyzing potential responses to climate responses to those astronomical cycles, as described below.


When not performing hypothesis tests for the presence of astronomical forcing, we recommend BIGSTACK$_{mixed}$, BIGSTACK$_{mixedP}$, and BIGSTACK$_{mixedA}$ for stratigraphic alignment because they use the maximum amount of age information – speleothem-based absolute dates from 0-654 ka and tuning to an astronomically forced, bipolar ice model for the rest. The regional stacks will work best for estimating ages of Atlantic and Pacific records by stratigraphic alignment, while

BIGSTACK$_{mixed}$ is the default option for other oceans or estimating the global mean changes. When estimating global mean benthic $\delta^{18}$O change with the global BIGSTACK (on any age model), users should be aware that, like LR04 and ProbStack, it is not volume-weighted. Of the cores in BIGSTACK$_{mixed}$, 56% are from the Atlantic and 36% are from the Pacific.

Paleoceanographic studies frequently seek to test whether responses to astronomical forcing can be detected in a record. When

considering the last 650 kyr, all of the new stacks in this study (except BIGSTACK$_{LR04}$) are suitable for testing the presence of eccentricity, obliquity (except 41-kyr cycles) or precession power as well as the precession phase between the NH and SH signals, because none have been tuned using insolation, eccentricity, obliquity (except 41-kyr cycles), or precession (Fig. 2). Because BIGSTACK$_{auto}$ and BIGSTACK$_{magrev}$ use the same $\delta^{18}$O values on different age models, they provide a consistent framework to test the hypothesis that a record >650 ka (aligned to the stacks) contains astronomical signals. For example, if a

hypothesis test for the presence of 41-kyr cyclicity fails when the record is aligned to BIGSTACK$_{magrev}$ but passes when aligned to BIGSTACK$_{auto}$, the hypothesis that the record contains a 41-kyr obliquity response is dependent upon astronomical tuning and should therefore be rejected. On the other hand, if a hypothesis test for obliquity passes when aligned to BIGSTACK$_{magrev}$, the hypothesis that the record contains an obliquity response likely can be accepted. Beyond 650 ka, the hand-tuned "mixed" stacks are tuned to a signal that includes all the major Milankovitch cycles, while BIGSTACK$_{auto}$ is minimally tuned to ~41

kyr obliquity power. Therefore, if precession or eccentricity frequencies can be detected when a certain record is aligned to BIGSTACK$_{auto}$ (or BIGSTACK$_{magrev}$), the hypothesis that it responds to precession or eccentricity forcing can be accepted. On the other hand, if precession or eccentricity frequencies can be detected when a certain record is aligned to the hand-tuned stacks but not when aligned to BIGSTACK$_{auto}$, the hypothesis that it contains power in precession or eccentricity could be an artifact of tuning to the ice volume model. Furthermore, BIGSTACK$_{auto}$ and BIGSTACK$_{magrev}$ can be used to study phase

relationships related to precession, obliquity, and eccentricity forcing that are free of circular reasoning related to tuning.





## 6 Conclusions

Constructing an age model for Pleistocene sediments presents various challenges. Aligning a benthic foraminiferal $\delta^{18}O$ record to a target is a common way to construct an age model beyond the range of radiocarbon. The LR04 stack (Lisiecki and Raymo, 2005) is frequently used as a target for such purposes, but as a global stack, it does not account for regional divergences in benthic $\delta^{18}O$. The LR04 stack also does not include many newer records and age model constraints published over the past 20 years. Furthermore, existing regional stacks do not cover the entire Pleistocene. Here we present three stacks – a global stack with three different age models and separate Atlantic and Pacific stacks, BIGSTACK$_{mixedA}$, and BIGSTACK$_{mixedP}$. The regional stacks and the global BIGSTACK$_{mixed}$ use age models transferred from absolutely dated archives for 0-654 ka and an astronomically tuned ice model for the rest of the Pleistocene. Because the Atlantic and Pacific regional stacks differ in some features, they can improve the stratigraphic alignment of sediments from these two intensely studied ocean basins. BIGSTACK$_{mixed}$ is the recommended option for other oceans or estimating the global mean changes. The stacks with mixed age models should be the best choice for most age model construction because of their flexibility and the incorporation of both absolute and tuned age information. BIGSTACK$_{magrev}$, which is based on geomagnetic reversal ages and a sediment compaction correction, is useful for assessing climatic responses to astronomical forcing with less risk of circular reasoning. Alternatively, BIGSTACK$_{auto}$ algorithmically concentrates the ~41 kyr obliquity power and is constructed with minimal subjective interventions. The combined use of multiple versions of the stacks provided by this study can be used for testing whether a record of interest contains astronomical signals and the impact of different types of assumptions on Pleistocene age estimates.

## Data Availability

- Data in support of this study have been uploaded to a Zenodo repository. The repository will be made publicly available once accepted for publication.
- The BIGSTACKs published in this study can be interactively explored at this webpage: https://yz3062.github.io/htmls/BIGSTACK_bokeh.html

## Author contribution

YZ and LEL: Conceptualization, Investigation, Writing – original draft, Writing – review & editing. SRM: Investigation, Writing – review & editing. TL and CL: Writing – review & editing.

## Competing interests

The authors declare that they have no conflict of interest.



## Acknowledgments

- We thank Doug Wilson for helpful discussions on geomagnetic timescale ages. Funding to support this research was provided by the Heising-Simons Foundation (Grant 2021- 2799 to L.E.L, and Grant 2021-2797 to S.R.M.) and by the National Science Foundation (OCE-2410906 to T.L.).
- Use was made of computational facilities purchased with funds from the National Science Foundation (CNS-1725797) and administered by the Center for Scientific Computing (CSC). The CSC is supported by the California NanoSystems Institute and the Materials Research Science and Engineering Center (MRSEC; NSF DMR2308708)
at UC Santa Barbara. We also acknowledge high-performance computing support from Casper (doi: https://doi.org/10.5065/D6RX99HX) provided by NCAR's Computational and Information Systems Laboratory, sponsored by the National Science Foundation.

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
