# Peer review of "Global and regional Pleistocene benthic $\delta^{18}O$ stacks with a comparison of different age modeling strategies"

_EGUsphere, 2025_

## Author Comment (AC2)

Response to comments from reviewers

Key
– Reviewers' comments
– Authors' response

**Referee #1**

**Review of "Global and regional Pleistocene benthic δ18O stacks with a comparison of different age modeling strategies" by Zhou et al.**

**General comments**

The authors propose an update to the LR04 benthic d18O stack (Lisiecki & Raymo 2005) that is widely used to construct age models of marine sediments in paleoceanographic studies. This update, which relies on an extended number of sites (224 cores), includes one global stack presented on 3 different chronologies and two regional stacks (Atlantic, Pacific). The manuscript describes the construction of these stacks and their chronologies. It also discusses the age uncertainties associated with them and the applications for which each stack is best suited.

There is absolutely no doubt that the new stacks proposed by the authors are of high scientific importance and will be widely adopted by the paleoceanographic community. Overall, the manuscript is well written and illustrated. It would have been helpful to include more information on the sites selected for each chronological iteration, as well as a more detailed discussion on specific aspects, which are detailed below.

Response #1 – We thank the reviewer for recognizing the scientific importance of our project. Below, we address the comments as fully as we can in order to take advantage of the input and improve our manuscript.

**Specific comments**

1. The Methods section starts directly with the description of stacking steps. I think a new subsection should be added first to better describe the marine cores selected for the construction of each stack and the related criteria: number of cores, proportions of new sites vs. cores already included in previous stacks, name of cores, their location (including water-depth), their references, mean temporal resolution of records, which proxies are available for each core apart from benthic d18O (magnetic data, SST, others?). Right now, this information is partly disseminated throughout the text, partly present in the supplementary data or missing. I think presenting an overview of selected cores and criteria in the main text and explicitly referring to the supplementary data for more information would be helpful.

Response #2 – We agree with the reviewer that the Methods section should begin with a new subsection describing the marine cores used for the stacks. This subsection will include all the information the reviewer mentioned: the number of cores, proportions of new sites vs. cores

already included in previous stacks, names of cores, their locations (including water depth), their references, and the mean temporal resolution of the records. For information on cores that are too long to include in the main manuscript, such as their locations, we will provide an aggregated statistical overview. We will also discuss the geomagnetic reversal and biostratigraphic age information when available. However, we note that our stack construction does not use SST data. While SST data are frequently measured in marine sediments and can similarly inform glacial/interglacial trends as benthic $\delta^{18}O$, compiling all the SST records associated with over 200 cores is outside the scope of this study and best suited to a follow-up investigation. See Response #8 for further clarification of this point.

2. I am surprised that the list of geomagnetic reversal and excursions selected for BIGSTACK$_{magrev}$ (Table 1) does not include recent well-dated magnetic events such as the Mono-Lake, Laschamps and Blake events. Reading the discussion in lines 411-416, I guess it is because only geomagnetic reversal/excursions dates derived from astronomical tuning are included. First, I would place this statement earlier in the text, in the Methods section. Second, I think one advantage of using magnetic reversal/excursions is also to benefit from absolute and relatively accurate dating from Ar/Ar. I find it really unfortunate that the attempt to use Ar/Ar dating did not succeed (lines 414-417). I would encourage the authors to further detail and illustrate this attempt. And if difficulties arise in the interval around 1.9-2.1 Ma only, why not reduce the coverage of the stack to the most recent 1.9 Ma or simply omit the period 1.9-2.1 Ma in the produced stack? More discussion on this issue would be helpful.

Response #3 – We agree with the reviewer that using Ar/Ar dated geomagnetic reversal/excursion ages is highly preferable to using astronomically tuned ages. We revisited the BIGSTACK$_{magrev}$ construction process and found a way to use Ar/Ar dated magnetic reversal/excursion ages that still allows the eTimeOpt procedure to work. In brief, we found two cores (ODP 1082 and 1083) that only have data in the oldest segment of the stack (~1.9 Ma and older), which may have biased the magrev age model towards younger ages. After removing these two cores, the auto-tuning improved significantly. See below the comparison of the old and new magrev and auto stacks (Figs. 1 and 2), as well as the time-frequency analysis results from the multi-taper method evolutive harmonic analysis before and after tuning for the new auto stack (Fig. 3). Note that even though ODP 1082 and 1083 only have data in the oldest segment of the stack, their removal impacts younger segments of the stack because of the interpolation between geomagnetic reversals/excursions.

In the revised manuscript, we plan to switch to using the new versions of BIGSTACK$_{magrev}$ and BIGSTACK$_{auto}$. We will include a comparison of the old and new magrev age models (and their tuning results) in the supplementary material to illustrate the age models' sensitivity to estimating magnetic reversal ages in these two different ways (tuned ages vs Ar/Ar ages).

We have found identifications of the Laschamp and Blake events in ODP 983, 984, and 1089 in the literature (Channell et al., 1997; Channell, 1999; Stoner et al., 2003) and incorporated the Ar/Ar ages of these events in the construction of BIGSTACK$_{magrev}$ (Singer, 2014). Mono-Lake is recorded in only one marine sedimentary site (ODP 919) off east Greenland (Channell et al.,

[Figure]

Figure 1. A comparison of the current BIGSTACK$_{magrev}$ (red) and the new version of BIGSTACK$_{magrev}$ (blue) that uses Ar/Ar-dated reversal/excursion ages.

2020), and the core from this site does not have an abundance of benthic foraminifera (Channell, 2006).

Given the high scientific importance of the produced stacks, I think that their relative comparisons are too brief and insufficiently discussed. Because stacks are plotted on different Y-axes in available figures (Figures 6, 7, etc), I find it very difficult to compare the amplitude and the timing of the stacks. I think it is particularly important for the comparison of the global BIGSTACKmixed stack to the regional Atlantic and Pacific stacks. I would invite the authors to plot these stacks on the same Y-axes and develop the discussion on their similarities and differences. Also, I would find it interesting to reinforce the discussion by adding a comparison, in addition to LR04, of the new stacks to ProbStack and HN23 stack, which is currently absent in the manuscript. Finally, more discussion on the reasons and the impacts of a higher smoothing effect in the BIGSTACKs compared to LR04 (lines 402-404) would be helpful. Indeed, I have the impression that the five BIGSTACKs are highly smoothed compared to LR04 (see https://yz3062.github.io/htmls/BIGSTACK_bokeh.html). I already found LR04 quite smoothed when using it to derive age models of sediment cores for the last glacial cycles. I anticipate the

[Figure]

Figure 2. A comparison of the current BIGSTACK$_{auto}$ (red) and the new version of BIGSTACK$_{auto}$ (blue) that derives from a BIGSTACK$_{magrev}$ that uses Ar/Ar-dated reversal/excursion ages (blue curve in Fig. 1). Note that the new version of BIGSTACK$_{auto}$ (blue) uses a longer window size of 500 kyrs in eTimeOpt, which shortens the resulting stack slightly more.

high smoothing of BIGSTACKs may be a major drawback when using them as references to produce marine chronologies. I am wondering to which extent this smoothing could be minimized during the stacking approach and would expect this smoothing to be better discussed in the text, in particular highlighting this potential limitation in scientific applications.

Response #4 – We wholly agree with the reviewer that the comparison between BIGMACS$_{mixed}$, BIGMACS$_{mixedA}$, and BIGMACS$_{mixedP}$ can be improved, both visually and in the text. We will also add a discussion of our stacks with ProbStack and H23NA. Lastly, we agree with the reviewer that the smoothness of the new stacks should be examined more carefully. This point was also brought up by Reviewer 2. We will highlight the degree of smoothing of the new

[Figure]

Figure 3. The time-frequency analysis results from the multi-taper method evolutive harmonic analysis, before and after tuning for a new version of BIGMACS$_{auto}$ that uses an input stack that uses Ar/Ar-dated reversal/excursion ages.

stacks, discuss why the stacks are so smooth (due to BIGMACS's use of Gaussian Process Regression), and assess the implications of this smoothing for their scientific applications. We plan to add a statement to the paper explaining that this stack is specifically geared towards orbital-scale age models, and that we advise against using it for millennial-scale stratigraphy of the most recent glacial cycles. For high-resolution regional and global stratigraphy of the last glacial cycle, we will point readers to Lisiecki and Stern (2016). We will also add supplemental figures comparing the Lisiecki and Stern (2016) Deep North Atlantic stack with our BIGSTACK$_{mixedA}$ and comparing the Lisiecki and Stern (2016) volume-weighted global stack with our BIGSTACK$_{mixed}$ to illustrate the difference.

3. I am not convinced by the temporal lag of 1 ka that is applied to derive the Pacific stack compared to the Atlantic one. Even though this issue is already discussed in lines 466-477, I am wondering whether this approach is the most robust. As already mentioned in

the text (line 468), I think the Pacific-Atlantic lag is larger than 1 ka during deglaciations. Have the authors considered using a larger Pacific-Atlantic lag during deglaciations, instead of or in addition to a constant throughout the investigated time period?

Response #5 – We agree with the reviewer's assessment that applying a constant temporal lag of 1 kyr to the Pacific stack compared to the Atlantic is overly simplistic. We plan to increase the Pacific lag to 2 kyr during the middle of each termination, then ramp it down to 1 kyr at the interglacial peak and the glacial maximum. The 2-kyr lag during the middle of each termination is supported by a previous study that estimated the average Pacific lag during Late Pleistocene terminations to be 1600 yr (Lisiecki and Raymo, 2009).

**Technical comments**

Line 17: The reference Ahn et al. 2017 is missing in the reference list.

Response #6 – We thank the reviewer for the careful reading of our manuscript. The reference is there, but it is hard to see because it seems to be part of the previous item in the bibliography due to the lack of indentation.

Lines 68-70: This section describes the "strengths and weaknesses" of chronological approaches. For radiocarbon dating, I would mention in these lines the difficulty to correctly estimate past marine age reservoir values.

Response #7 – Agreed.

Line 153: please clarify what is meant with "multiproxy age models". Do the authors mean age models that use several proxy-records from the same core?

Response #8 – BIGMACS allows the simultaneous use of radiocarbon and benthic $\delta^{18}$O to create an age model. It also allows the use of additional age control points, such as geomagnetic reversals/excursions and biostratigraphy. However, it cannot currently stratigraphically align multiple proxy time series such as benthic $\delta^{18}$O and SST. We agree that this term can be ambiguous and will clarify that, by 'multiproxy,' we mainly mean benthic $\delta^{18}$O and age control points (e.g., radiocarbon ages, magnetic reversals/excursions, and/or biostratigraphic events).

Lines 227-230: I think this information should arrive earlier in the text.

Response #9 – Agreed.

Line 305: I think the tie-points that have been defined to manually align BIGMACSmagrev to the different targets to produce the mixed BIGSTACK should be provided in the supplementary data files. It may already be the case, but I did not find them to check.

Response #10 – Agreed. We will provide that information as a supplementary data file.

Figure 5: please add the colour code in the figure's caption so that the reader knows which colored curve corresponds to which stacks.

Response #11 – Will do.

Figures 6, 7, 9, 10: please add the name of the stacks directly onto the figure (e.g. along the Y-axes or next the record).

Response #12 – Will do.

Figure 8: similarly please indicate directly onto the figure which age difference is plotted in panels a, b, and c.

Response #13 – Will do.

Supplementary Data file 10: it would be helpful to specify the content of this supplementary data. To which stacking step or chronology do ages given in this excel file correspond for all cores?

Response #14 – This file contains details of input records for $BIGSTACK_{mixedP}$ and $BIGSTACK_{mixedA}$. We will add a new "README" tab at the beginning that explains the file's contents.

---

## Author Response (AR1)

Response to comments from reviewers

Key
  – Reviewers' comments
  – Authors' response

**Referee #1**

**Review of "Global and regional Pleistocene benthic δ18O stacks with a comparison of different age modeling strategies" by Zhou et al.**

**General comments**

The authors propose an update to the LR04 benthic d18O stack (Lisiecki & Raymo 2005) that is widely used to construct age models of marine sediments in paleoceanographic studies. This update, which relies on an extended number of sites (224 cores), includes one global stack presented on 3 different chronologies and two regional stacks (Atlantic, Pacific). The manuscript describes the construction of these stacks and their chronologies. It also discusses the age uncertainties associated with them and the applications for which each stack is best suited.

There is absolutely no doubt that the new stacks proposed by the authors are of high scientific importance and will be widely adopted by the paleoceanographic community. Overall, the manuscript is well written and illustrated. It would have been helpful to include more information on the sites selected for each chronological iteration, as well as a more detailed discussion on specific aspects, which are detailed below.

Response #1 – We thank the reviewer for recognizing the scientific importance of our project. Below, we address the comments as fully as we can in order to take advantage of the input and improve our manuscript.

**Specific comments**

1. The Methods section starts directly with the description of stacking steps. I think a new subsection should be added first to better describe the marine cores selected for the construction of each stack and the related criteria: number of cores, proportions of new sites vs. cores already included in previous stacks, name of cores, their location (including water-depth), their references, mean temporal resolution of records, which proxies are available for each core apart from benthic d18O (magnetic data, SST, others?). Right now, this information is partly disseminated throughout the text, partly present in the supplementary data or missing. I think presenting an overview of selected cores and criteria in the main text and explicitly referring to the supplementary data for more information would be helpful.

Response #2 – We have revised the Methods section to begin with a new subsection describing the marine cores used for the stacks (L156) (Note: all line numbers refer to the revised manuscript with track changes turned on). This subsection includes all the information the

reviewer mentioned: the number of cores, proportions of new sites vs. cores already included in previous stacks, names of cores, their locations (including water depth), their references, and the mean temporal resolution of the records. For information on cores that are too long to include in the main manuscript, such as their locations, we have provided an aggregated statistical overview. We have also discussed the geomagnetic reversal and biostratigraphic age information when available. However, we note that our stack construction does not use SST data. While SST data are frequently measured in marine sediments and can similarly inform glacial/interglacial trends as benthic $\delta^{18}O$, compiling all the SST records associated with over 200 cores is outside the scope of this study and best suited to a follow-up investigation. See Response #8 for further clarification of this point.

2. I am surprised that the list of geomagnetic reversal and excursions selected for BIGSTACK$_{magrev}$ (Table 1) does not include recent well-dated magnetic events such as the Mono-Lake, Laschamps and Blake events. Reading the discussion in lines 411-416, I guess it is because only geomagnetic reversal/excursions dates derived from astronomical tuning are included. First, I would place this statement earlier in the text, in the Methods section. Second, I think one advantage of using magnetic reversal/excursions is also to benefit from absolute and relatively accurate dating from Ar/Ar. I find it really unfortunate that the attempt to use Ar/Ar dating did not succeed (lines 414-417). I would encourage the authors to further detail and illustrate this attempt. And if difficulties arise in the interval around 1.9-2.1 Ma only, why not reduce the coverage of the stack to the most recent 1.9 Ma or simply omit the period 1.9-2.1 Ma in the produced stack? More discussion on this issue would be helpful.

Response #3 – We agree with the reviewer that using Ar/Ar dated geomagnetic reversal/excursion ages is highly preferable to using astronomically tuned ages. We revisited the BIGSTACK$_{magrev}$ construction process and found a way to use Ar/Ar dated magnetic reversal/excursion ages that still allows the eTimeOpt procedure to work. In brief, we found two cores (ODP 1082 and 1083) that only have data in the oldest segment of the stack (~1.9 Ma and older), which may have biased the magrev age model towards younger ages. After removing these two cores, the auto-tuning improved significantly. See below the comparison of the old and new magrev and auto stacks (Response Figs. 1 and 2), as well as the time-frequency analysis results from the multi-taper method evolutive harmonic analysis before and after tuning for the new auto stack (Response Fig. 3). Note that even though ODP 1082 and 1083 only have data in the oldest segment of the stack, their removal impacts younger segments of the stack because of the interpolation between geomagnetic reversals/excursions.

In the revised manuscript, we have switched to using the new versions of BIGSTACK$_{magrev}$ and BIGSTACK$_{auto}$. We have included a comparison of the old and new magrev age models (and their tuning results) in the supplementary material (Fig. S4 and S5) to illustrate the age models' sensitivity to estimating magnetic reversal ages in these two different ways (tuned ages vs Ar/Ar ages).

We have found identifications of the Laschamp and Blake events in ODP 983, 984, and 1089 in the literature (Channell et al., 1997; Channell, 1999; Stoner et al., 2003) and incorporated the Ar/Ar ages of these events in the construction of BIGSTACK$_{magrev}$ (Singer, 2014). Mono-Lake is

[Figure]

Figure 1. A comparison of the current BIGSTACK_{magrev} (red) and the new version of BIGSTACK_{magrev} (blue) that uses Ar/Ar-dated reversal/excursion ages.

recorded in only one marine sedimentary site (ODP 919) off east Greenland (Channell et al., 2020), and the core from this site does not have an abundance of benthic foraminifera (Channell, 2006).

Given the high scientific importance of the produced stacks, I think that their relative comparisons are too brief and insufficiently discussed. Because stacks are plotted on different Y-axes in available figures (Figures 6, 7, etc), I find it very difficult to compare the amplitude and the timing of the stacks. I think it is particularly important for the comparison of the global BIGSTACKmixed stack to the regional Atlantic and Pacific stacks. I would invite the authors to plot these stacks on the same Y-axes and develop the discussion on their similarities and differences. Also, I would find it interesting to reinforce the discussion by adding a comparison, in addition to LR04, of the new stacks to ProbStack and HN23 stack, which is currently absent in the manuscript. Finally, more discussion on the reasons and the impacts of a higher smoothing effect in the BIGSTACKs compared to LR04 (lines 402-404) would be helpful. Indeed, I have the impression that the five BIGSTACKs are highly smoothed compared to LR04 (see https://yz3062.github.io/htmls/BIGSTACK_bokeh.html). I already found LR04 quite smoothed

[Figure]

Figure 2. A comparison of the current BIGSTACK$_{auto}$ (red) and the new version of BIGSTACK$_{auto}$ (blue) that derives from a BIGSTACK$_{magrev}$ that uses Ar/Ar-dated reversal/excursion ages (blue curve in Response Fig. 1). Note that the new version of BIGSTACK$_{auto}$ (blue) uses a longer window size of 500 kyrs in eTimeOpt, which shortens the resulting stack slightly more.

when using it to derive age models of sediment cores for the last glacial cycles. I anticipate the high smoothing of BIGSTACKs may be a major drawback when using them as references to produce marine chronologies. I am wondering to which extent this smoothing could be minimized during the stacking approach and would expect this smoothing to be better discussed in the text, in particular highlighting this potential limitation in scientific applications.

Response #4 – We wholly agree with the reviewer that the comparison between BIGSTACK$_{mixed}$, BIGSTACK$_{mixedA}$, and BIGSTACK$_{mixedP}$ can be improved, both visually and in the text. We have added a figure putting the three BIGSTACKs on the same y-axis (Fig. 9) and a discussion of their differences (L494). We have also added a figure comparing BIGSTACK$_{mixed}$ to LR04, ProbStack, and H23NA (Fig. 13). We have also added a discussion of our stacks with the previous stacks (L801). Lastly, we agree with the reviewer that the smoothness of the new stacks should be examined more carefully. This point was also brought up by Reviewer 2. We

[Figure]

Figure 3. The time-frequency analysis results from the multi-taper method evolutive harmonic analysis, before and after tuning for a new version of BIGMACS$_{auto}$ that uses an input stack that uses Ar/Ar-dated reversal/excursion ages.

have highlighted the degree of smoothing of the new stacks, discussed why the stacks are so smooth (due to BIGMACS's use of Gaussian Process Regression), and assessed the implications of this smoothing for their scientific applications (L812). We have added a statement to the paper explaining that this stack is specifically geared towards orbital-scale age models, and that we advise against using it for millennial-scale stratigraphy of the most recent glacial cycles. For high-resolution regional and global stratigraphy of the last glacial cycle, we point readers to Lisiecki and Stern (2016). We have also added supplemental figures comparing the Lisiecki and Stern (2016) Deep North Atlantic stack with our BIGSTACK$_{mixedA}$ and comparing the Lisiecki and Stern (2016) volume-weighted global stack with our BIGSTACK$_{mixed}$ to illustrate the difference (Fig. S2 and S6).

3. I am not convinced by the temporal lag of 1 ka that is applied to derive the Pacific stack compared to the Atlantic one. Even though this issue is already discussed in lines 466-477, I am wondering whether this approach is the most robust. As already mentioned in

the text (line 468), I think the Pacific-Atlantic lag is larger than 1 ka during deglaciations. Have the authors considered using a larger Pacific-Atlantic lag during deglaciations, instead of or in addition to a constant throughout the investigated time period?

Response #5 – We agree with the reviewer's assessment that applying a constant temporal lag of 1 kyr to the Pacific stack compared to the Atlantic is overly simplistic. We have increased the Pacific lag to 2 kyr during the middle of each termination, then ramp it down to 1 kyr at the interglacial peak and the glacial maximum (Response Fig. 4). The 2-kyr lag during the middle of each termination is supported by a previous study that estimated the average Pacific lag during Late Pleistocene terminations to be 1600 yr (Lisiecki and Raymo, 2009). BIGSTACK$_{mixedP}$ has been updated in the manuscript.

[Figure]

Figure 4. Comparison of the BIGSTACK$_{mixedP}$ before and after adding an additional 1-kyr lag during deglaciations compared to the Atlantic stack. The orange arrows point out the subtle effects of this added delay.

**Technical comments**

Line 17: The reference Ahn et al. 2017 is missing in the reference list.

Response #6 – We thank the reviewer for the careful reading of our manuscript. The reference is there, but it is hard to see because it seems to be part of the previous item in the bibliography due to the lack of indentation.

Lines 68-70: This section describes the "strengths and weaknesses" of chronological approaches. For radiocarbon dating, I would mention in these lines the difficulty to correctly estimate past marine age reservoir values.

Response #7 – Done (L74).

Line 153: please clarify what is meant with "multiproxy age models". Do the authors mean age models that use several proxy-records from the same core?

Response #8 – BIGMACS allows the simultaneous use of radiocarbon and benthic $\delta^{18}O$ to create an age model. It also allows the use of additional age control points, such as geomagnetic reversals/excursions and biostratigraphy. However, it cannot currently stratigraphically align multiple proxy time series such as benthic $\delta^{18}O$ and SST. We agree that this term can be ambiguous and have clarified that, by 'multiproxy,' we mainly mean benthic $\delta^{18}O$ and age control points (e.g., radiocarbon ages, magnetic reversals/excursions, and/or biostratigraphic events) (L184).

Lines 227-230: I think this information should arrive earlier in the text.

Response #9 – Agreed.

Line 305: I think the tie-points that have been defined to manually align BIGMACSmagrev to the different targets to produce the mixed BIGSTACK should be provided in the supplementary data files. It may already be the case, but I did not find them to check.

Response #10 – Agreed. We have provided that information as a supplementary data file (Supplementary Data File 11).

Figure 5: please add the colour code in the figure's caption so that the reader knows which colored curve corresponds to which stacks.

Response #11 – Done.

Figures 6, 7, 9, 10: please add the name of the stacks directly onto the figure (e.g. along the Y-axes or next the record).

Response #12 – Done.

Figure 8: similarly please indicate directly onto the figure which age difference is plotted in panels a, b, and c.

Response #13 – Done.

Supplementary Data file 10: it would be helpful to specify the content of this supplementary data. To which stacking step or chronology do ages given in this excel file correspond for all cores?

Response #14 – This file contains details of input records for BIGSTACK$_{mixedP}$ and BIGSTACK$_{mixedA}$. We have added a new "README" tab at the beginning that explains the file's contents.

References

Channell, J. E. T.: Geomagnetic paleointensity and directional secular variation at Ocean Drilling Program (ODP) Site 984 (Bjorn Drift) since 500 ka: Comparisons with ODP Site 983 (Gardar Drift), Journal of Geophysical Research: Solid Earth, 104, 22937–22951, https://doi.org/10.1029/1999JB900223, 1999.

Channell, J. E. T.: Late Brunhes polarity excursions (Mono Lake, Laschamp, Iceland Basin and Pringle Falls) recorded at ODP Site 919 (Irminger Basin), Earth and Planetary Science Letters, 244, 378–393, https://doi.org/10.1016/j.epsl.2006.01.021, 2006.

Channell, J. E. T., Hodell, D. A., and Lehman, B.: Relative geomagnetic paleointensity and $\delta 18O$ at ODP Site 983 (Gardar Drift, North Atlantic) since 350 ka, Earth and Planetary Science Letters, 153, 103–118, https://doi.org/10.1016/S0012-821X(97)00164-7, 1997.

Channell, J. E. T., Singer, B. S., and Jicha, B. R.: Timing of Quaternary geomagnetic reversals and excursions in volcanic and sedimentary archives, Quaternary Science Reviews, 228, 106114, https://doi.org/10.1016/j.quascirev.2019.106114, 2020.

Lisiecki, L. E. and Raymo, M. E.: Diachronous benthic $\delta^{18}O$ responses during late Pleistocene terminations, Paleoceanography, 24, https://doi.org/10.1029/2009PA001732, 2009.

Lisiecki, L. E. and Stern, J. V.: Regional and global benthic $\delta^{18}O$ stacks for the last glacial cycle, Paleoceanography, 31, 1368–1394, https://doi.org/10.1002/2016PA003002, 2016.

Singer, B. S.: A Quaternary geomagnetic instability time scale, Quaternary Geochronology, 21, 29–52, https://doi.org/10.1016/j.quageo.2013.10.003, 2014.

Stoner, J. S., Channell, J. E. T., Hodell, D. A., and Charles, C. D.: A ~580 kyr paleomagnetic record from the sub-Antarctic South Atlantic (Ocean Drilling Program Site 1089), Journal of Geophysical Research: Solid Earth, 108, https://doi.org/10.1029/2001JB001390, 2003.

Key
– Reviewers' comments
– Authors' response

**Referee #2**

This manuscript presents an ambitious and valuable update to the venerable LR04 stack by compiling a much larger dataset and introducing new regional (Atlantic and Pacific) stacks. The study's goals are highly relevant for Quaternary paleoclimate and stratigraphy. The paper is generally well written and organized, and the figures are informative. In its current form, however, several aspects need clarification or further development to maximize the manuscript's clarity, transparency, and utility. In particular, the methods and rationale for certain chronological choices should be explained in more detail, comparisons to previous work should be expanded, and some figures and data descriptions require improvements for clarity. I recommend major revisions to address the issues raised above before the manuscript is accepted for publication in EGUsphere/Geochronology.

Comments:

1. The manuscript would benefit from a clearer description of the cores and data that went into each stack. Currently, information on the 224 cores (sources, locations, water depths, new vs. previously published records, resolution, etc.) is scattered or only in the supplement. I recommend adding a summary table or a dedicated subsection in Methods describing the core selection criteria and dataset characteristics.

Response #15 – We agree with the reviewer. This is also brought up by Reviewer 1. See Response #2 for how we have revised the manuscript to better describe the cores in our compilation.

2. Given the prominence of LR04 and ProbStack, the manuscript should more fully compare the new stacks to these earlier benchmarks. The text does note that the new global stack includes ~50% more data than ProbStack, but there is little quantitative or visual comparison in the paper. I recommend adding some discussion (and possibly a figure or table) comparing the new global stack in terms of mean trends, variability, and age offsets.

Response #16 – We agree with the reviewer. We have added a figure comparing the new stacks to LR04 and ProbStack (Fig. 13). See Response #4 for more details.

3. Can anything be done to retain more signal variance? While a smooth stack yields a high signal-to-noise ratio for orbital-scale trends, the manuscript should caution that some climatic signals (e.g. abrupt events, smaller excursions) might be blunted. I encourage the authors to

highlight this limitation so that users of BIGSTACK understand the potential need to cross-check against individual records for fine-scale events.

Response #17 – We agree that the smoothing of the BIGSTACKs is a limitation that should be better communicated to the readers. We have explained in the revised manuscript why the stacks are so smooth (due to BIGMACS's use of Gaussian Process Regression), and discuss the implications of this smoothing for their scientific applications (L812) (Note: all line numbers refer to the revised manuscript with track changes turned on). See Response #4 for more details.

4. Figs. 5–7: Please add clear labels or legends identifying each colored curve.

Response #18 – Done.